# GENERALIZATION ERROR MINIMIZED DEEP LEARNING

## ABSTRACT

Despite the vast applications and rapid development of deep learning (DL), understanding and improving the generalization ability of deep neural networks (DNNs) remains a fundamental challenge. To tackle this challenge, in this paper, we first establish a novel bias-variance decomposition framework to analyze the generalization error of DNNs. Based on our new generalization error formula, we then present a new form of DL dubbed generalization error minimized (GEM) DL by jointly minimizing the conventional optimization target and an analytical proxy for the generalization error. Extensive experimental results show that in comparison with DNNs trained within the standard DL, GEM DNNs have smaller generalization errors and better generalization ability, thereby improving DNN prediction accuracy. Notably, GEM DL can increase prediction accuracy by as much as 13.19% on ImageNet in the presence of data distribution shift between training and testing.

## 1 INTRODUCTION

In the past decade, deep neural networks (DNNs) have demonstrated impressive success in a wide range of applications. In spite of this, the overfitting problem remains prevalent and significant, greatly affecting the generalization performance of DNNs. Broadly speaking, overfitting is a phenomenon that DNNs perform good or even perfect on training data, but much worse on new, unseen data. In other words, DNNs that suffer from overfitting have poor generalization ability.

In the literature, the overfitting issue of DNNs has been extensively studied. Theoretical frameworks, such as bias-variance tradeoff (Geman et al., 1992) and generalization bounds (Kawaguchi et al., 2017) shed some light on the generalization behavior of DNNs and provide practitioners with qualitative guidance to train DNNs with better generalization. However, it's rare or difficult for them to be directly applied in learning algorithms to prevent overfitting. On the other hand, there are also a myriad of empirical works that manage to reduce overfitting and train DNNs with improved generalization performance. These empirical works, in general, lack strong theoretical foundations, resulting in limited universality and explainability. To overcome these drawbacks, the purpose of this paper is to bridge the gap between these two lines of research.

To begin with, we define the generalization error of a learned DNN as the expectation of the squared difference between training performance and testing performance of the DNN. Using this definition, we then establish a novel bias-variance decomposition framework within which the generalization error of the learned DNN can be decomposed into the sum of three terms: (1) the expectation of conditional testing variance, (2) the expectation of conditional training variance, and (3) the expectation of bias between training and testing. In parallel, we also define the conditional generalization error of a learned DNN given its weight vector, which can be decomposed similarly into the sum of conditional testing variance, conditional training variance, and bias between training and testing. The decomposition formula has clear implications on how to design an effective learning algorithm. It suggests that to achieve a small generalization error, we should search for a model that minimizes these three terms jointly with the conventional training objective such as the empirical negative log-likelihood risk in the case of classification. Among the three terms, it is difficult to represent the conditional training variance analytically. To make such a joint optimization problem tractable, we further propose analytical proxies for the generalization error and the conditional generalization error. Upper bound the expectation of conditional training variance by the uncondi-

tional training variance. By demonstrating empirically that the unconditional training variance is negligible in comparison with the generalization error, it turns out that the analytical proxies are close approximations to the generalization error and the conditional generalization error. Based on the analytical proxy of the conditional generalization error, we then modify the conventional deep learning (DL) by jointly optimizing the conventional training loss and the analytical proxy, yielding a new form of DL dubbed generalization error minimized (GEM[1]) DL. To verify the effectiveness of GEM DL, we conducted extensive experiments for image classification on CIFAR-100 (Krizhevsky et al., 2009) and ImageNet (Deng et al., 2009). It is shown, by experiments, that for a variety of DNN architectures, in comparison with DNNs trained within the conventional DL, GEM DNNs trained within GEM DL indeed have smaller generalization errors and better generalization ability, achieving consistent gains in prediction accuracy. It's worth noting that GEM DL can outperform the convention DL by up to 13.19% in accuracy on ImageNet when there is a data distribution shift between training and testing. Moreover, the superior performance of GEM DL over the conventional DL is demonstrated in few-shot and imbalanced data scenarios as well.

The major contributions of this paper are summarized as follows:

- We give new definitions of generalization error and conditional generalization error of a learned DNN, and establish novel bias-variance decomposition formulas for them, offering new insights into the generalization behavior of DNNs.

- We present analytical proxies (i.e., close approximations) for the generalization error and the conditional generalization error.

- Based on the new bias-variance decomposition and analytical proxies, we develop a new training framework dubbed GEM DL, which jointly minimizes the conventional training loss and the analytical proxy for the conditional generalization error.

- The superior performance of GEM DL over the conventional DL is further confirmed by extensive experiments on CIFAR-100 and ImageNet for a variety of DNN architectures and different application scenarios including JPEG compression, Gaussian blurring, few-shot learning and imbalanced data scenarios.

## 2 RELATED WORK

Extensive efforts have been made in recent years to understand generalization in DL. Among them are a body of principled mathematical works on deriving generalization bounds (Kawaguchi et al., 2017; Xu & Raginsky, 2017; Jakubovitz et al., 2019; Jiang et al., 2019; Neu et al., 2021), which upper bound the population risk by terms related to the training data and the model's properties. While significant progress has been made, many of these established bounds remain loose or even vacuous, offering limited insight into the remarkable generalization abilities of neural networks observed in practice (Gastpar et al., 2023). Moreover, although generalization bounds are primarily developed to explain generalization behavior and provide generalization guarantees, they often fall short in being applied to improve generalization in DL empirically.

Also related to this above line of theoretical works are those works on bias-variance tradeoff (Geman et al., 1992; Domingos, 2000). It is well known that the expected error on an unseen test sample can be decomposed into the bias, which measures the discrepancy between the model class and the data distribution, and the variance, which measures the variability of the model given the randomness involved in training. Conventional wisdom suggests that as a model's complexity—often measured by the number of its parameters—increases, its variance increases while its bias decreases (Geman et al., 1992). This belief has long influenced model selection practices, advocating for a model that is neither too simple nor too complex in order to strike the optimal balance between bias and variance, thereby minimizing generalization error. However, this view has been called into question by numerous recent works (Zhang et al., 2017a; Novak et al., 2018; Neal et al., 2018; Belkin et al., 2019; Yang et al., 2020), which present evidence that modern neural networks often benefit from increased capacity, contradicting the classical bias-variance tradeoff. Controversy aside, like generalization bounds, the classical bias-variance decomposition primarily serves as theoretical guidance

---

[1]Throughout the paper, GEM will stand for either "generalization error minimized" or "generalization error minimization".

for model selection but does not directly contribute to training a given DNN model with reduced generalization error.

In contrast, in this paper, we analyze, for any given learned DNN, its generalization error and conditional generalization error defined as the unconditional and conditional expectation of the squared difference between training performance and testing performance of that DNN. We establish a new form of bias-variance decomposition, which can be applied directly to train a given DNN model with reduced generalization error and better generalization performance by jointly minimizing the conventional training loss and the analytical proxy for the conditional generalization error. Its practicality is similar to that of existing empirical methods aimed at reducing overfitting and improving generalization of DNNs, including weight decay (Krogh & Hertz, 1991), early stopping (Morgan & Bourlard, 1989), dropout (Srivastava et al., 2014), label smoothing (Szegedy et al., 2016), confidence penalty (Pereyra et al., 2017), and data augmentation such as Cutout (DeVries & Taylor, 2017), Mixup (Zhang et al., 2017b) and Cutmix (Yun et al., 2019). Compared to these empirical methods, our approach is grounded in solid mathematical foundations, as it directly leverages the new bias-variance decomposition formula. More importantly, our method is orthogonal to existing techniques in modern DL designed to improve DNN generalization, as it can further enhance DNN performance when combined with those methods.

## 3 GENERALIZATION ERROR ANALYSIS

### 3.1 DEFINITION OF GENERALIZATION ERROR

Let $(X, Y)$ be a pair of random variables, the distribution of which governs training data, where $X$ represents an input to a DNN, and $Y$ is the ground truth label of $X$. Let $D = \{(x_1, y_1), (x_2, y_2), \ldots, (x_n, y_n)\}$ be a training set with size $n$, where $\{(x_i, y_i)\}_{i=1}^n$ is a sequence of independent copies of $(X, Y)$. Let $f_\theta$ denote a DNN architecture parameterized by a weight vector $\theta$, which, given $\theta$, outputs $f_\theta(x)$ in response to an input $x$. When a learning algorithm endowed with a loss function $\mathcal{L}(f_\theta(x), y)$ is applied to $D$ and the DNN architecture, a learned model $f_{\hat{\theta}}$ is generated, where $\hat{\theta}$ is the learned weight vector. In general, $\hat{\theta}$ is a solution or an approximate solution to

$$\min_\theta \frac{1}{n} \sum_{i=1}^n \mathcal{L}(f_\theta(x_i), y_i) \approx \min_\theta \mathbb{E}[\mathcal{L}(f_\theta(X), Y)], \tag{1}$$

where "$\approx$" is valid with high probability by the law of large numbers. Once the trained model $f_{\hat{\theta}}$ is obtained, we can measure its training performance by

$$\Omega(D, \hat{\theta}) = \frac{1}{n} \sum_{i=1}^n \mathcal{L}(f_{\hat{\theta}}(x_i), y_i). $$

Let $(U, V)$ be another pair of random variables, the distribution of which governs testing data, where $U$ represents an input to a DNN, and $V$ is the ground truth label of $U$. In general, $(X, Y)$ and $(U, V)$ may or may not have the same probability distribution. Let $T = \{(u_1, v_1), (u_2, v_2), \ldots, (u_m, v_m)\}$ be a testing set with size $m$, where $\{(u_j, v_j)\}_{j=1}^m$ is a sequence of independent copies of $(U, V)$. Applying the trained model $f_{\hat{\theta}}$ to $T$, one can measure its testing performance by

$$\Omega(T, \hat{\theta}) = \frac{1}{m} \sum_{j=1}^m \mathcal{L}(f_{\hat{\theta}}(u_j), v_j). $$

Note that since $D$ is random, so is $f_{\hat{\theta}}$. Define the generalization error of $f_{\hat{\theta}}$ as

$$\Gamma = \mathbb{E}\left[\Omega(D, \hat{\theta}) - \Omega(T, \hat{\theta})\right]^2, \tag{2}$$

where the expectation is taken with respect to $D$, $T$, and all other random elements involved in the training process. Throughout this paper, $T$ is assumed to be independent of $D$ and the training process. Given any $\theta$, let

$$\Gamma(\theta) = \mathbb{E}\left[\Omega(D, \hat{\theta}) - \Omega(T, \hat{\theta}) \Big| \hat{\theta} = \theta\right]^2. \tag{3}$$

In view of the law of total expectation, it follows that $\Gamma = \mathbb{E}[\Gamma(\hat{\theta})]$. Subsequently, we refer to $\Gamma(\theta)$ as the conditional generalization error of $f_{\hat{\theta}}$ given $\hat{\theta} = \theta$. The smaller $\Gamma$ is, the better $f_{\hat{\theta}}$ generalizes. Thus, to attain $f_{\hat{\theta}}$ with better generalization, one should take $\Gamma$ into consideration during training. In order to implement this idea, we first analyze $\Gamma$ through a nice bias-variance decomposition.

## 3.2 DECOMPOSITION OF GENERALIZATION ERROR

We begin with the preparation of some quantities which come handy in the subsequent decomposition. Note that since $\hat{\theta}$ depends only on $D$ and the training process, it follows that $T$ and $\hat{\theta}$ are independent. Given $\hat{\theta} = \theta$, let

$$K(\theta) = \mathbb{E}\left[\Omega(T, \hat{\theta})\,\Big|\,\hat{\theta} = \theta\right] = \mathbb{E}_{U,V}\left[\mathcal{L}(f_\theta(U), V)\right], \tag{4}$$

where $\mathbb{E}_{U,V}$ denotes the expectation with respect to $(U, V)$. Similarly, given $\hat{\theta} = \theta$, let

$$J(\theta) = \mathbb{E}\left[\Omega(D, \hat{\theta})\,\Big|\,\hat{\theta} = \theta\right]. \tag{5}$$

Note that $K(\theta)$ and $J(\theta)$ are the conditionally expected testing performance and the conditionally expected training performance given $\hat{\theta} = \theta$, respectively.

**Theorem 1.** *Let $\Gamma$ and $\Gamma(\theta)$ be the generalization error and conditional generalization error defined in (2) and (3). Then the following hold:*

$$\Gamma(\theta) = \mathrm{Var}(\Omega(T, \hat{\theta})\,\big|\,\hat{\theta} = \theta) + \mathrm{Var}(\Omega(D, \hat{\theta})\,\big|\,\hat{\theta} = \theta) + [J(\theta) - K(\theta)]^2, \tag{6}$$

*and*

$$\Gamma = \mathbb{E}\left[\mathrm{Var}(\Omega(T, \hat{\theta})\,\big|\,\hat{\theta})\right] + \mathbb{E}\left[\mathrm{Var}(\Omega(D, \hat{\theta})\,\big|\,\hat{\theta})\right] + \mathbb{E}\left[J(\hat{\theta}) - K(\hat{\theta})\right]^2, \tag{7}$$

*where*

$$\mathrm{Var}(\Omega(T, \hat{\theta})|\hat{\theta}) = \mathbb{E}_T\left[(\Omega(T, \hat{\theta}) - K(\hat{\theta}))^2\,\Big|\,\hat{\theta}\right]$$

*is the conditional variance of $\Omega(T, \hat{\theta})$ given $\hat{\theta}$ and*

$$\mathrm{Var}(\Omega(D, \hat{\theta})|\hat{\theta}) = \mathbb{E}_{D|\hat{\theta}}\left[(\Omega(D, \hat{\theta}) - J(\hat{\theta}))^2\,\Big|\,\hat{\theta}\right]$$

*is the conditional variance of $\Omega(D, \hat{\theta})$ given $\hat{\theta}$.*

*Proof.* By assumption, $T$ is independent of $D$ and $\hat{\theta}$. For any $\theta$, it can be verified that

$$\Gamma(\theta) = \mathbb{E}\left[(\Omega(D, \hat{\theta}) - \Omega(T, \hat{\theta}))^2\,\Big|\,\hat{\theta} = \theta\right]$$

$$= \mathbb{E}_{D|\hat{\theta}=\theta} E_T\left[(\Omega(D, \hat{\theta}) - \Omega(T, \hat{\theta}))^2\,\Big|\,\hat{\theta} = \theta\right] \tag{8}$$

$$= \mathbb{E}_{D|\hat{\theta}=\theta}\left[\mathrm{Var}(\Omega(T, \hat{\theta})|\hat{\theta} = \theta) + (\Omega(D, \hat{\theta}) - K(\hat{\theta}))^2\,\Big|\,\hat{\theta} = \theta\right] \tag{9}$$

$$= \mathrm{Var}(\Omega(T, \hat{\theta})|\hat{\theta} = \theta) + \mathbb{E}_{D|\hat{\theta}=\theta}\left[(\Omega(D, \hat{\theta}) - K(\hat{\theta}))^2\,\Big|\,\hat{\theta} = \theta\right]$$

$$= \mathrm{Var}(\Omega(T, \hat{\theta})|\hat{\theta} = \theta) + \mathrm{Var}(\Omega(D, \hat{\theta})|\hat{\theta} = \theta) + [J(\theta) - K(\theta)]^2, \tag{10}$$

where (8) follows from the fact that $T$ is independent of $(D, \hat{\theta})$, (9) follows (4), and (10) is due to (5). This completes the proof of (6). The decomposition (7) follows from (6) and the law of total expectation. $\qquad\square$

From Theorem 1, the generalization error $\Gamma$ is decomposed into three meaningful terms, namely (1) the expectation of conditional testing variance $\mathbb{E}[\mathrm{Var}(\Omega(T, \hat{\theta})|\hat{\theta})]$, (2) the expectation of conditional training variance $\mathbb{E}[\mathrm{Var}(\Omega(D, \hat{\theta})|\hat{\theta})]$, and (3) the expectation of bias between training and testing $\mathbb{E}[J(\hat{\theta}) - K(\hat{\theta})]^2$. In the next section, we will leverage this result to propose a new form of deep learning.

## 4 GENERALIZATION ERROR MINIMIZED DL

### 4.1 PROXIES FOR $\Gamma(\theta)$ AND $\Gamma$

In order to attain a DNN model $f_{\hat{\theta}}$ with small generalization error, it follows from Theorem 1 that a desirable learning algorithm should generate $\hat{\theta}$ so that both $\Gamma(\hat{\theta})$ and the conventional training loss $\mathbb{E}[\mathcal{L}(f_{\hat{\theta}}(X), Y)]$ are small. Expand the last term in (6), and rewrite $\Gamma(\hat{\theta})$ as follow

$$\Gamma(\hat{\theta}) = \mathrm{Var}(\Omega(T, \hat{\theta})|\hat{\theta}) + K^2(\hat{\theta}) + \mathrm{Var}(\Omega(D, \hat{\theta})|\hat{\theta}) + J(\hat{\theta})[J(\hat{\theta}) - 2K(\hat{\theta})]. \tag{11}$$

Since $T$ and $\hat{\theta}$ are independent, the first two terms on the right side of (11) have neat analytical expressions, which will be clear later. The fourth term therein is generally small since $J(\hat{\theta})$ is related to the conventional training loss $\mathbb{E}[\mathcal{L}(f_{\hat{\theta}}(X), Y)]$, and the training process aims to minimize it. Hence, the fourth term on the right side of (11) can be ignored. The third term on the right side of (11), however, is problematic; it is intractable due to the intervolving nature between $D$ and $\hat{\theta}$. To tackle this problem, we go back to (7) and examine how large the expectation of conditional training variance $\mathbb{E}[\mathrm{Var}(\Omega(D, \hat{\theta})|\hat{\theta})]$ could be. At this point, we invoke the following proposition:

**Proposition 1.** *Let* $\mathrm{Var}(\Omega(D, \hat{\theta}))$ *be the unconditional training variance. Then we have the following upper bound*

$$\mathbb{E}\left[\mathrm{Var}(\Omega(D, \hat{\theta})\,\middle|\,\hat{\theta})\right] \leq \mathrm{Var}(\Omega(D, \hat{\theta})). \tag{12}$$

*Proof.* This is a simple consequence of the law of total variance. $\qquad\square$

Intuitively, the unconditional training variance $\mathrm{Var}(\Omega(D, \hat{\theta}))$ should be small, given the fact that $\Omega(D, \hat{\theta})$ is always small as long as $\hat{\theta}$ is effectively optimized on $D$. This is indeed confirmed later by our numerical analysis in A.1, which shows empirically that $\mathrm{Var}(\Omega(D, \hat{\theta})) \ll \Gamma$ in practice. As a result, it follows from (12) that the expectation of conditional training variance $\mathbb{E}[\mathrm{Var}(\Omega(D, \hat{\theta})|\hat{\theta})]$ is even more negligible. This, combined with Markov inequality, further implies that with high probability, $\mathrm{Var}(\Omega(D, \hat{\theta})|\hat{\theta})$ is negligible as well and hence can be ignored from (11). Therefore, both $\Gamma(\hat{\theta})$ and $\Gamma$ can be approximated, respectively, by

$$\Gamma(\hat{\theta}) \approx \hat{\Gamma}(\hat{\theta}) = \mathrm{Var}(\Omega(T, \hat{\theta})|\hat{\theta}) + K^2(\hat{\theta}) \tag{13}$$

and

$$\Gamma \approx \hat{\Gamma} = \mathbb{E}[\mathrm{Var}(\Omega(T, \hat{\theta})|\hat{\theta})] + \mathbb{E}[K^2(\hat{\theta})]. \tag{14}$$

Hereafter, $\hat{\Gamma}(\hat{\theta})$ and $\hat{\Gamma}$ are referred to as proxies for $\Gamma(\hat{\theta})$ and $\Gamma$, respectively.

Since $T$ is independent of $\hat{\theta}$, the proxy $\hat{\Gamma}(\theta)$, given $\hat{\theta} = \theta$, depends only on the distribution of $(U, V)$, but not on the testing dataset $T$ itself, and actually has a neat analytical expression, as shown below:

$$\begin{aligned}
\hat{\Gamma}(\theta) &= \mathrm{Var}(\Omega(T, \hat{\theta})|\hat{\theta} = \theta) + K^2(\theta) \\
&= \mathrm{Var}(\Omega(T, \theta)) + K^2(\theta) \\
&= \frac{1}{m}\left(\mathbb{E}[\mathcal{L}(f_\theta(U), V)]^2 - K^2(\theta)\right) + K^2(\theta) \\
&= \frac{1}{m}\mathbb{E}[\mathcal{L}(f_\theta(U), V)]^2 + \frac{m-1}{m}K^2(\theta) \\
&= \frac{1}{m}\mathbb{E}[\mathcal{L}(f_\theta(U), V)]^2 + \frac{m-1}{m}(\mathbb{E}[\mathcal{L}(f_\theta(U), V)])^2 \tag{15}
\end{aligned}$$

where (15) is due to (4).

Two special cases are particularly interesting: Case 1 where $(U, V)$ and $(X, Y)$ have the same distribution; Case 2 where $(U, V)$ and $(X, Y)$ have different distributions, but the distribution of $(U, V)$ can be obtained from $(X, Y)$ through common signal processing such as JPEG compression, Gaussian blurring, etc. In these two special cases, $\hat{\Gamma}(\theta)$ can be rewritten as

$$\hat{\Gamma}(\theta) = \frac{1}{m}\mathbb{E}[\mathcal{L}(f_\theta(X), Y)]^2 + \frac{m-1}{m}(\mathbb{E}[\mathcal{L}(f_\theta(X), Y)])^2 \tag{16}$$

in Case 1, and as

$$\hat{\Gamma}(\theta) = \frac{1}{m}\mathbb{E}[\mathcal{L}(f_\theta(\hat{X}), Y)]^2 + \frac{m-1}{m}(\mathbb{E}[\mathcal{L}(f_\theta(\hat{X}), Y)])^2 \tag{17}$$

in Case 2, where $\hat{X}$ is a processed version of $X$. Each expected value in (16) and (17) can be approximated by its respective empirical mean over a mini-batch of the training dataset $D$.

## 4.2 GEM DEEP LEARNING

Based on our discussions above, instead of minimizing the conventional training loss

$$\mathcal{L}_{ERM}(\theta) = \mathbb{E}[\mathcal{L}(f_\theta(X), Y)] \tag{18}$$

where ERM stands for empirical risk minimization, our proposed GEM DL minimizes both $\mathcal{L}_{ERM}(\theta)$ and the generalization error proxy $\hat{\Gamma}(\theta)$ jointly. Define

$$\mathcal{L}_{GEM}(\theta) = \mathbb{E}[\mathcal{L}(f_\theta(X), Y)] + \lambda\mathbb{E}[\mathcal{L}(f_\theta(U), V)]^2 + \beta(\mathbb{E}[\mathcal{L}(f_\theta(U), V)])^2 \tag{19}$$

where $(U, V)$ is replaced by $(X, Y)$ in Case 1, and by $(\hat{X}, Y)$ in Case 2, and where $\lambda \geq 0$ and $\beta \geq 0$ are two hyperparameters. In other words, in GEM DL we solve the following optimization problem instead

$$\min_\theta \mathcal{L}_{GEM}(\theta) \tag{20}$$

where the hyperparameter $\beta$ is introduced to give us more flexibility without being restricted to the relationship $\beta = (m-1)\lambda$ as shown in (15) to (17). Note that on the right side of (19), the second expectation is the second moment, and the term after $\beta$ is the squared mean.

By minimizing $\mathcal{L}_{GEM}(\theta)$, we essentially guide the training process to search for a DNN model with small generalization error while minimizing the empirical risk, so as to prevent overfitting and improve generalization.

## 4.3 APPLICATION TO CLASSIFICATION

Despite its universality, in this paper we apply GEM DL to the classification task only, given the popularity of the latter in DL applications. In multiclass classification, a DNN $f_\theta(x) = p_\theta(\cdot|x)$ is mathematically a mapping from an input $x \in \mathcal{X}$ to a probability vector $p_\theta(\cdot|x) \in \Delta_C$, where $C$ is the number of all possible classes. Conventionally, the loss function for classification is the negative log-likelihood (NLL) loss, i.e., $\mathcal{L}(f_\theta(x), y) = -\log p_\theta(y|x)$, where $y$ is the ground truth label corresponding to $x$. Therefore, for the classification task, we have

$$\mathcal{L}_{GEM}(\theta) = \mathbb{E}[-\log p_\theta(Y|X)] + \lambda\mathbb{E}[-\log p_\theta(V|U)]^2 + \beta(\mathbb{E}[-\log p_\theta(V|U)])^2 \tag{21}$$

where $(U, V)$ is replaced by $(X, Y)$ in Case 1, and by $(\hat{X}, Y)$ in Case 2.

When label smoothing (LS) Szegedy et al. (2016) is applied to regularize $\mathcal{L}(f_\theta(x), y) = -\log p_\theta(y|x)$, we simply replace the the first term on the right side of (21) by

$$\mathbb{E}[-(1-\alpha)\log p_\theta(Y|X) + \alpha D_{\mathrm{KL}}(u\|p_\theta(\cdot|X))], \tag{22}$$

yielding

$$\mathcal{L}_{GEM}(\theta) = \mathbb{E}[-(1-\alpha)\log p_\theta(Y|X) + \alpha D_{\mathrm{KL}}(u\|p_\theta(\cdot|X))]$$
$$+ \lambda\mathbb{E}[-\log p_\theta(V|U)]^2 + \beta(\mathbb{E}[-\log p_\theta(V|U)])^2 \tag{23}$$

where $(U, V)$ is replaced by $(X, Y)$ in Case 1, and by $(\hat{X}, Y)$ in Case 2, $u$ is a uniform distribution over all $C$ possible classes, $\alpha$ controls the strength of the smoothing effect, and $D_{\mathrm{KL}}(u\|p_\theta(\cdot|X))$ is the divergence between $u$ and $p_\theta(\cdot|X)$.

The same principle adopted in (22) and (23) will be applied to handle other regularization terms for ERM as well, which will not affect the terms related to $\hat{\Gamma}(\theta)$ in $\mathcal{L}_{GEM}(\theta)$, in order to guarantee the orthogonality of GEM to the existing DL pipeline. As a result, GEM is completely plug-and-play, meaning that there's no need to tweak anything in the existing training pipeline except for a slight modification to the objective function, which introduces negligible extra complexity.

## 5 EXPERIMENTS

### 5.1 EXPERIMENTAL SETTINGS

We benchmark GEM on two popular image classification datasets, namely CIFAR-100 (Krizhevsky et al., 2009) and ImageNet (Deng et al., 2009).

**CIFAR-100**: For all experiments on this dataset, including those for GEM and other compared methods, we follow the training recipe from CRD (Tian et al., 2019) without any adjustment. The evaluated model architectures are MobileNetV2 (Sandler et al., 2018), ShuffleNetV2 (Ma et al., 2018), WideResNet (Zagoruyko & Komodakis, 2016), ResNet (He et al., 2016) and VGG (Simonyan & Zisserman, 2014). On this dataset, we set $(\lambda, \beta) = (0.005, 0.05)$ for $\mathcal{L}_{GEM}$, and the same hyperparameters are shared across all models.

**ImageNet**: For all experiments on this dataset, including those for GEM and other compared methods, we employ the standard training recipes from PyTorch (Paszke et al., 2019) without any adjustment. The evaluated model architectures are ShuffleNetV2, SqueezeNet (Iandola et al., 2016) and ResNet. On this dataset, we set $(\lambda, \beta) = (0.002, 0.01)$ for $\mathcal{L}_{GEM}$, and the same hyperparameters are shared across all models.

## 5.2 STANDARD TASKS

In this subsection, all experimental results correspond to Case 1 mentioned in Section 4.1, where $(U, V)$ and $(X, Y)$ follow the same distribution, as is typically assumed for datasets like CIFAR-100 and ImageNet. Results for Case 2 will be presented in the next subsection.

**Results on CIFAR-100**. The performance of GEM is shown in Table 1. We compare it with the baseline ERM and another competitive method dubbed DOM (Lin et al., 2024) which also targets to address the overfitting problem. Across all the six tested models, GEM consistently provides significant gains over the compared methods, demonstrating its effectiveness in improving the generalization of DNNs.

Table 1: Top-1 test accuracy (%) on CIFAR-100. The baseline ERM results are from Tian et al. (2019), while the results of DOM and GEM are averaged over 3 runs and reported with the standard deviation. Note that we highlight the best results in bold, and $\Delta$ stands for the performance improvement over ERM.

| Method | MobileNetV2 | ShuffleNetV2 | WRN-40-2 | resnet32 | resnet20 | vgg8 |
|---|---|---|---|---|---|---|
| ERM | 64.60 | 71.82 | 75.61 | 71.14 | 69.06 | 70.36 |
| DOM | $65.07 \pm 0.23$ | $72.62 \pm 0.30$ | $76.19 \pm 0.40$ | $71.25 \pm 0.36$ | $69.16 \pm 0.23$ | $70.43 \pm 0.24$ |
| $\Delta$ | +0.47 | +0.80 | +0.58 | +0.11 | +0.10 | +0.07 |
| **GEM** | $\mathbf{65.99} \pm 0.59$ | $\mathbf{73.17} \pm 0.30$ | $\mathbf{76.93} \pm 0.31$ | $\mathbf{71.95} \pm 0.30$ | $\mathbf{69.84} \pm 0.18$ | $\mathbf{71.04} \pm 0.18$ |
| $\Delta$ | +1.39 | +1.35 | +1.32 | +0.81 | +0.78 | +0.68 |

**Results on ImageNet**. In Table 2, we demonstrate the performance of GEM compared to ERM and DOM. It's shown that GEM consistently improves the performance over ERM for all tested models, while DOM fails to show gain on any model. Thus, the effectiveness of GEM is further verified on large-scale dataset, where overfitting has already been largely mitigated by the abundance of training data and strong training recipes.

Table 2: Top-1 test accuracy (%) on ImageNet. The ERM and DOM results are based on our reproduction following the standard Pytorch training recipes. More reimplementation details about DOM can be found in A.4.

| Method | ShuffleNetV2 | SqueezeNetV1.1 | ResNet18 | ResNet34 |
|---|---|---|---|---|
| ERM | 59.17 | 57.95 | 69.76 | 73.31 |
| DOM | 58.67 | 56.78 | 68.99 | 72.73 |
| $\Delta$ | -0.50 | -1.17 | -0.77 | -0.58 |
| **GEM** | **59.99** | **58.33** | **70.09** | **73.51** |
| $\Delta$ | +0.82 | +0.38 | +0.33 | +0.20 |

## 5.3 ADDITIONAL TASKS

**GEM in Case 2**. To validate the effectiveness of GEM in Case 2 as mentioned in Section 4.1, we consider two types of common image processing: JPEG compression and Gaussian blurring.

JPEG compression is of particular interest due to its widespread use in real-world applications. However, DNNs trained with high quality images often generalize badly to low quality JPEG compressed images (Yang et al., 2021b; Zheng et al., 2023; Salamah et al., 2024). To address this issue, GEM in Case 2 can be applied. Specifically, we train ResNet18 on the ImageNet training set using $\mathcal{L}_{GEM}$ in Case 2, where $\hat{X}$ is a JPEG compressed version of $X$ with some JPEG quality factor $q^2$. We then evaluate the trained model on JPEG compressed versions of the ImageNet validation set with varying values of $q$. For all experiments, we use the TorchJPEG (Ehrlich et al., 2020) library to compress images. Results are shown in Fig. 1(a). As noted in the literature, the model trained with high quality images using ERM indeed generalizes poorly to low quality images, resulting in over 26% accuracy degradation at $q = 10$. In contrast, when GEM in Case 2 is applied, where $\hat{X}$ is a JPEG compressed version of $X$ with $q = 30$, the trained model performs comparably to the ERM baseline at the highest quality level while consistently and increasingly outperforming the baseline as $q$ decreases. In the case where $\hat{X}$ is a JPEG compressed version of $X$ with $q = 10$, GEM sacrifices a bit of performance at high quality levels, but in turn obtains a substantial improvement at low quality levels, achieving 13.19% gain over the baseline at $q = 10$.

Gaussian blurring is another important case to examine, as it simulates real-world scenarios where a camera is not properly focused on the object of interest. Using a setup similar to that of JPEG compression, we train ResNet18 on the ImageNet training set using $\mathcal{L}_{GEM}$ in Case 2, where $\hat{X}$ is a Gaussian blurred version of $X$ with a fixed Gaussian kernel size of 9 and some standard deviation $\sigma$. We then evaluate the trained model on Gaussian blurred versions of the ImageNet validation set with the same kernel size and varying values of $\sigma$. As shown in Fig. 1(b), the results for Gaussian blurring follow a similar pattern to those for JPEG compression. When $\hat{X}$ is produced by Gaussian blurring with $\sigma = 1$, GEM outperforms the baseline ERM on all blurred validation sets, and roughly maintains the performance on the raw validation set. On the other hand, when $\hat{X}$ is produced by Gaussian blurring with $\sigma = 3$, GEM sacrifices more at low blurring levels, but in turn obtains a significant improvement at high blurring levels, achieving 6.56% gain over the baseline at $\sigma = 3$.

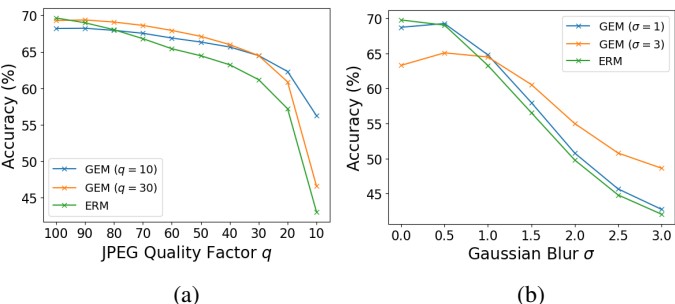

(a)  (b)

Figure 1: Top-1 test accuracy (%) comparison on ImageNet in the case where $(U, V)$ and $(X, Y)$ have different distributions, but the distribution of $(U, V)$ can be obtained from $(X, Y)$ through common signal processing such as (a) JPEG compression, and (b) Gaussian blurring.

**Few-shot learning**. In the few-shot scenario, DNNs are trained with limited amount of data. Following Yang et al. (2021a), we sample class-balanced subsets from the CIFAR-100 training set to serve as small training sets used in few-shot learning. To be specific, we collect four subsets of CIFAR-100 training set by retaining 10%, 25%, 50% and 75% of the training images. Then, we train MobileNetV2 with GEM and ERM on these balanced subsets using the same training recipe for CIFAR-100 as mentioned in Section 5.1, and evaluate the trained models on the complete CIFAR-100 testing set. For fair comparison, the same subsets are used for both GEM and ERM. Note that the few-shot scenario falls into Case 1 where $(U, V)$ and $(X, Y)$ have the same distribution, so we apply GEM in Case 1 accordingly.

As shown in Fig. 2, GEM consistently outperforms ERM with a large margin, and the performance improvement gradually increases as the training set becomes smaller and smaller, showing the exceptional ability of GEM to mitigate overfitting on small datasets. This trend in performance gain

---

[2]The quality factor $q$ of JPEG is an integer ranging from 1 to 100. A lower $q$ indicates more compression and consequently lower quality of the compressed image.

makes intuitive sense as DNNs suffer more from overfitting with smaller training sets, so that GEM can achieve more gain by reducing generalization error. Note that $\lambda$ and $\beta$ in few-shot learning are increased to 0.01 and 0.2 to better handle the increase in overfitting.

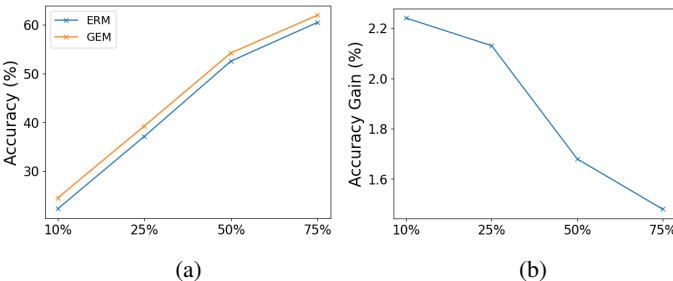

(a)                                         (b)

Figure 2: (a) Top-1 test accuracy (%) comparison on CIFAR-100 in few-shot scenario using different percentages of training samples. Results are averaged over 3 runs. (b) The accuracy gain achieved by GEM compared to the baseline ERM.

**Imbalanced dataset**. We also test the behavior of GEM on imbalanced datasets. Following Cao et al. (2019), we create imbalanced subsets of the CIFAR-100 training set by reducing the number of training examples per class and keep the testing set intact. Same as Cao et al. (2019), two modes of imbalance are considered, namely long-tailed imbalance (Cui et al., 2019) and step imbalance (Mateusz et al., 2018). Long-tailed imbalance results in an exponential decrease in the number of samples across different classes, while in the step imbalance setting, all minority classes have the same sample size, so do all frequent classes. In the case of step imbalance, half of the classes are minority classes, while the other half are frequent classes. Moreover, in both imbalance modes, an imbalance factor, defined as the ratio between sample sizes of the least frequent class and the most frequent class, is used to measure the degree of imbalance. Clearly, a smaller imbalance factor stands for higher degree of imbalance. In our evaluation, we consider four different imbalance factors: 0.01, 0.02, 0.05 and 0.1. Again, we train MobileNetV2 with GEM and ERM on these imbalanced subsets using the same training recipe for CIFAR-100 as mentioned in Section 5.1, and $(\lambda, \beta)$ values for GEM are the same as those used in few-shot learning. Note that there's a distribution shift between $(U, V)$ and $(X, Y)$ due to the class imbalance of the training set; however, this type of distribution shift cannot be characterized by any common signal processing over the data, and therefore the imbalance data scenario doesn't fall into either Case 1 or Case 2. Nevertheless, we still apply GEM in Case 1 for this task.

As shown in Fig. 3, in both imbalance modes, GEM outperforms ERM by a large margin under high imbalance factors, i.e., when the degree of imbalance is low, while the gain gradually diminishes with the presence of more imbalance. This trend in performance gain comes naturally as GEM in Case 1 doesn't take the distribution shift between training and testing into consideration. Therefore, the gain is becoming less and less with increased imbalance which induces increased distribution shift. Actually, it's quite surprising to see that GEM maintains its effectiveness in some cases where it's not specially designed for. A more general form of GEM that can better address class imbalance or the out-of-distribution (OOD) issue in broad sense is left to be explored in future work.

## 5.4 ANALYSIS AND DISCUSSION

**Generalization error curves.** In Fig. 4(a)-(c), we plot the generalization error curves for MobileNetV2, ShuffleNetV2 and WRN-40-2 trained on CIFAR-100. Two types of curve are depicted for (1) the squared difference between training NLL loss and test NLL loss, and (2) the difference between test error and training error, respectively. The former is a more direct reflection of the generalization error $\Gamma$ defined in this paper, while the latter relates more closely to the generalization performance of DNNs. In comparison with ERM, GEM consistently leads to less generalization error at the end of training in both types of generalization curve. Moreover, Fig. 4(d) shows the generalization error curves for MobileNetV2 trained on the 10% balanced subset of CIFAR-100 which is used in the few-shot learning scenario. In this case, GEM results in a significant reduction in generalization error compared to ERM, not just at the end of training but throughout the entire training

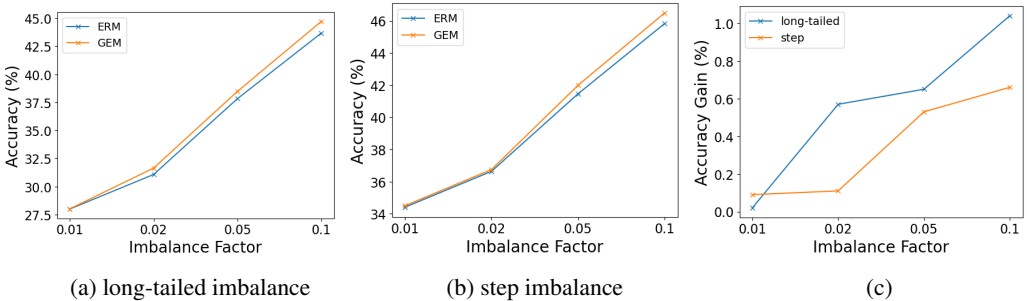

(a) long-tailed imbalance      (b) step imbalance      (c)

Figure 3: (a) Top-1 test accuracy (%) comparison on CIFAR-100 with long-tailed imbalance using different imbalance factors. Results are averaged over 3 runs. (b) Top-1 test accuracy (%) comparison on CIFAR-100 with step imbalance using different imbalance factors. Results are averaged over 3 runs. (c) The accuracy gain achieved by GEM compared to the baseline ERM under both imbalance modes.

process. Therefore, it's confirmed that GEM can indeed effectively reduce the generalization error and help DNNs generalize better.

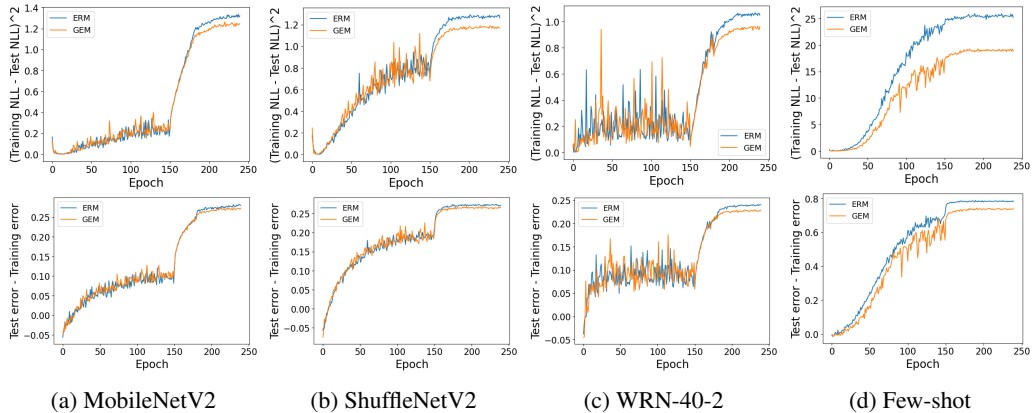

(a) MobileNetV2     (b) ShuffleNetV2     (c) WRN-40-2     (d) Few-shot

Figure 4: Generalization error curves for (a) MobileNetV2, (b) ShuffleNetV2, (c) WRN-40-2 trained on CIFAR-100, and (d) MobileNetV2 trained on CIFAR-100 under the 10% few-shot scenario. The figures in the first row depict the squared difference between training NLL loss and test NLL loss, while those in the second row depict the difference between test error and training error.

**Compatibility with the existing regularization**. For all conducted experiments, we didn't change anything in the default DL pipeline, where many regularization techniques including data augmentation like mixup, cutmix, etc., weight decay, and label smoothing have already been embedded to prevent overfitting and improve generalization. Nonetheless, GEM can still make further improvement. Therefore, it's confirmed empirically that GEM is compatible with the existing regularization, and can be easily applied in a plug-and-play manner.

## 6 CONCLUSION

The paper studies generalization error in DL and provides a novel learning algorithm to reduce it. A novel bias-variance decomposition is established to analyze the generalization error between training and testing, based on which we propose a new learning framework dubbed GEM by jointly minimizing the conventional training loss and an analytical proxy for the conditional generalization error. The proposed method is applied to classification and experimentally evaluated on popular image classification datasets, where it consistently and significantly mitigates overfitting and improves generalization performance of DNNs across a variety of scenarios. Overall, GEM is a simple yet effective approach to improving generalization, backed by solid theoretical support.

## REPRODUCIBILITY STATEMENT

Please refer to our source code provided in the Supplementary Material to reproduce our experimental results.

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

## A  APPENDIX

### A.1  NUMERICAL ANALYSIS ON $\mathrm{Var}(\Omega(D, \hat{\theta}))$ AND $\Gamma$

To justify the approximation introduced in Section 4.1, we conduct some numerical analysis on $\mathrm{Var}(\Omega(D, \hat{\theta}))$ and $\Gamma$ through experiments. Specifically, we train MobileNetV2, ShuffleNetV2 and WRN-40-2 using ERM on CIFAR-100, each with three runs. Then, we measure the squared difference between training NLL loss and test NLL loss at the end of training and average over three runs. The resulting value serves as an approximation of the generalization error $\Gamma$ defined in (2). Similarly, we approximate $\mathrm{Var}(\Omega(D, \hat{\theta}))$ by the sample variance of the training NLL loss at the end of training over three runs. In Table 3, we compare the resulting $\Gamma$ with $\mathrm{Var}(\Omega(D, \hat{\theta}))$, and it's clear that $\mathrm{Var}(\Omega(D, \hat{\theta}))$ is 4 to 8 orders of magnitude smaller than $\Gamma$. This supports our claim in Section 4.1 that $\mathrm{Var}(\Omega(D, \hat{\theta})) \ll \Gamma$ in practice, and consequently justifies the approximation made therein. Note that the training set $D$ is indeed random in this numerical analysis because random cropping and flipping are applied in the data augmentation.

Table 3: Comparison between $\Gamma$ and $\mathrm{Var}(\Omega(D, \hat{\theta}))$ on CIFAR-100.

|  | MobileNetV2 | ShuffleNetV2 | WRN-40-2 |
|---|---|---|---|
| $\Gamma$ | 1.33 | 1.21 | 1.03 |
| $\mathrm{Var}(\Omega(D, \hat{\theta}))$ | $5.07 \times 10^{-4}$ | $6.86 \times 10^{-8}$ | $1.90 \times 10^{-7}$ |

### A.2  IMPLEMENTATION OF GEM

In this section, we present Algorithm 1 as the pseudo-code of GEM in Case 1 in a Pytorch-like style.

---
**Algorithm 1** Pseudo-code of GEM in Case 1 in a Pytorch-like style.

---

```
# z: DNN output logits
# y: the ground truth label
# λ: the weight for the second moment
# β: the weight for the squared mean
# α: the amount of label smoothing

ce_ls = F.cross_entropy(z, y, label_smoothing=α)
ce = F.cross_entropy(z, y, reduction='none')
ce_mean = torch.mean(ce)
second_moment = torch.mean(ce**2)
squared_mean = ce_mean**2

# GEM
gem_loss = ce_ls + λ * second_moment + β * squared_mean
```
---

### A.3  GUIDANCE FOR HYPERPARAMETERS SELECTION

Qualitatively speaking, $\lambda$ in $\mathcal{L}_{GEM}$ should not be too large. At the beginning of training, DNNs are randomly initialized, so the loss $\mathcal{L}(f_\theta(x), y) = -\log p_\theta(y|x)$ is generally quite large, which is much greater than 1. Therefore, further squaring it will result in a even larger value. Consequently, if $\lambda$ is not small enough, DNNs will suffer from slow convergence or even gradient explosion at the beginning of training. As for $\beta$, it's suggested by (16) and 17 that $\beta$ should not be less than $\lambda$. Empirically, selecting $\beta$ to be about an order of magnitude greater than $\lambda$ often gives good results. Actually, as long as $\lambda$ and $\beta$ are appropriately small to avoid slow convergence and gradient explosion at the beginning of training, and $\beta$ is not less than $\lambda$, the experimental results are generally not sensitive to hyperparameters tuning.

### A.4 REIMPLEMENTATION DETAILS ABOUT DOM

The DOM method compared in this paper corresponds to $\text{DOM}_{\text{RE}}$ in the original paper. On CIFAR-100, we use 0.45 as the loss threshold for DOM as provided in the original paper. Also, the warm-up epoch is aligned with the first learning rate decay at the 150th epoch as suggested in the original paper.

As for ImageNet, since the original paper didn't evaluate DOM on ImageNet, we don't have a direct reference of hyperparmeters. Therefore, we adopt the adaptive loss threshold proposed in their Appendix G, setting it to 40% as suggested therein. Again, the warm-up epoch is aligned with the first learning rate decay for SqueezeNetV1.1, ResNet18 and ResNet34, while for ShuffleNetV2 which is trained using a cosine annealing learning rate scheduler for 600 epochs, we follow the convention in the original paper and align the warm-up epoch with the midpoint of training, i.e., the 300th epoch.

### A.5 MORE BENCHMARK COMPARISON IN CASE 1

Dropout and confidence penalty are two popular regularization techniques used to improve DNN generalization performance. In this section, we compare GEM in Case 1 with these two benchmarks empirically.

It's observed that the overconfidence of a DNN is a sign of overfitting and poor generalization (Szegedy et al., 2016). Thus, Pereyra et al. (2017) proposed to penalize confident predictions to improve DNN generalization performance. Since a confident prediction generally corresponds to $p_\theta(\cdot|X)$ with low entropy, they enforce confidence penalty (CP) by introducing a negative entropy regularizer into the learning objective, which is formulated as

$$\mathcal{L}_{CP}(\theta) = \mathbb{E}[-\log p_\theta(Y|X) - \eta H(p_\theta(\cdot|X))], \tag{24}$$

where $H(\cdot)$ is the entropy of a probability distribution, and $\eta$ controls the strength of the confidence penalty.

Under the experimental setting for CIFAR-100 reported in Section 5, we perform a grid search for $\eta$ over $\{0.1, 0.25, 0.5, 1.0, 1.5\}$ following the original paper and choose $\eta = 1$ as the optimal weight. The comparison between CP and GEM in Case 1 on CIFAR-100 is presented in Table 4. Clearly, GEM outperforms CP in general by a large margin.

Table 4: Top-1 test accuracy (%) on CIFAR-100.

| Method | MobileNetV2 | ShuffleNetV2 | WRN-40-2 | resnet32 | resnet20 | vgg8 |
|---|---|---|---|---|---|---|
| ERM | 64.60 | 71.82 | 75.61 | 71.14 | 69.06 | 70.36 |
| CP | $65.14 \pm 0.77$ | $\mathbf{73.46} \pm 0.25$ | $76.35 \pm 0.16$ | $71.21 \pm 0.25$ | $69.07 \pm 0.29$ | $70.97 \pm 0.25$ |
| $\Delta$ | +0.54 | +1.64 | +0.74 | +0.07 | +0.01 | +0.61 |
| **GEM** | $\mathbf{65.99} \pm 0.59$ | $73.17 \pm 0.30$ | $\mathbf{76.93} \pm 0.31$ | $\mathbf{71.95} \pm 0.30$ | $\mathbf{69.84} \pm 0.18$ | $\mathbf{71.04} \pm 0.18$ |
| $\Delta$ | +1.39 | +1.35 | +1.32 | +0.81 | +0.78 | +0.68 |

Dropout is yet another benchmark we compare with. Due to its long history, most modern DNNs have be designed with dropout layer in mind. However, for all DNN architectures we consider on CIFAR-100, only WideResNet (Zagoruyko & Komodakis, 2016) has incorporated dropout. Therefore, we only test dropout on this family of networks, where the designer finds dropout helpful. By varying the width and depth of WideResNet, we get three DNNs namely WRN-40-2, WRN-40-1, and WRN-16-2. For all of them, we set the dropout probability to be 0.3 following the original paper (Zagoruyko & Komodakis, 2016). The experimental setting is identical to the one reported in Section 5 and the hyperparameters for GEM also remain unchanged. The results in Table 5 show that dropout doesn't consistently improve the performance over the standard ERM, and always performs worse than GEM in Case 1.

### A.6 COMPARING WITH STABILITY TRAINING IN CASE 2

Stability Training (ST) (Zheng et al., 2016) is a method specially designed to improve DNN's generalization (robustness) to low quality images, whose loss is computed from both the training data

Table 5: Top-1 test accuracy (%) on CIFAR-100.

| Method | WRN-40-2 | WRN-40-1 | WRN-16-2 |
|--------|----------|----------|----------|
| ERM | 75.61 | 71.98 | 73.26 |
| Dropout | $76.68 \pm 0.21$ | $71.71 \pm 0.19$ | $73.33 \pm 0.20$ |
| $\Delta$ | +1.07 | -0.27 | +0.07 |
| **GEM** | $\mathbf{76.93} \pm 0.31$ | $\mathbf{72.09} \pm 0.28$ | $\mathbf{73.61} \pm 0.45$ |
| $\Delta$ | +1.32 | +0.11 | +0.35 |

$X$ and processed training data $\hat{X}$, thus providing a fair comparison with GEM in Case 2. The minimization objective of ST is

$$\mathcal{L}_{ST}(\theta) = \mathbb{E}[-\log p_\theta(Y|X) + \tau H(p_\theta(\cdot|X), p_\theta(\cdot|\hat{X}))], \qquad (25)$$

where $H(\cdot, \cdot)$ denotes the cross entropy between two probability distributions, and $\tau$ controls the strength of the stability term.

In Fig. 5, we compare GEM in Case 2 against ST under the JPEG compression scenario. Following the suggestion in the original paper, we set $\tau = 0.01$ for ST. Clearly, GEM outperforms ST in both cases where $\hat{X}$ is a JPEG compressed version of $X$ with $q = 10$ and $q = 30$.

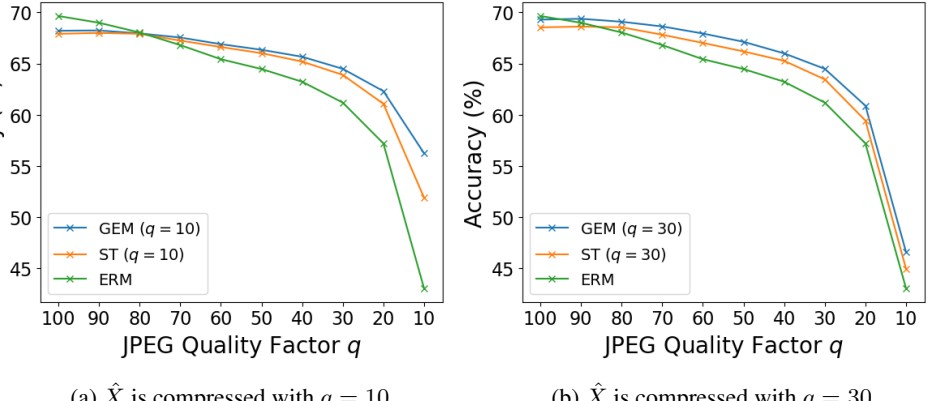

(a) $\hat{X}$ is compressed with $q = 10$.      (b) $\hat{X}$ is compressed with $q = 30$.

Figure 5: Top-1 test accuracy (%) comparison on ImageNet in the presence of JPEG compressed images.

### A.7 ANALYSIS ON SYNTHETIC DATASET

Spiral dataset is a classical synthetic dataset for binary classification. In order to analyze the difference between the models learned by GEM and ERM, we conduct a toy experiment on a synthetic spiral dataset and visualize the resulting decision boundaries.

We first create a spiral dataset with 2 classes, each consisting of 500 samples. Then, we split the dataset into training and testing sets with 4:1 ratio. A five-layer fully connected neural network, comprising four hidden layers each with a width of 1024, is constructed as the model to be trained. An Adam optimizer (Kingma, 2014) with learning rate 0.001 is used to train the model. When the model is trained using ERM, the test accuracy is 96.5%, and the decision boundary is visualized in Fig. 6(a). In the case of GEM, the test accuracy reaches 98.5%, and the learned decision boundary is visualized in Fig. 6(b). From the plots, we can clearly see that the decision boundary learned by ERM has some zigzag and dent (highlighted by red circles), while the decision boundary learned by GEM is much smoother and demonstrates a desirable central symmetry pattern. This clearly indicates that the model trained by GEM successfully captures the true data distribution, while the one trained using ERM overfits the noise.

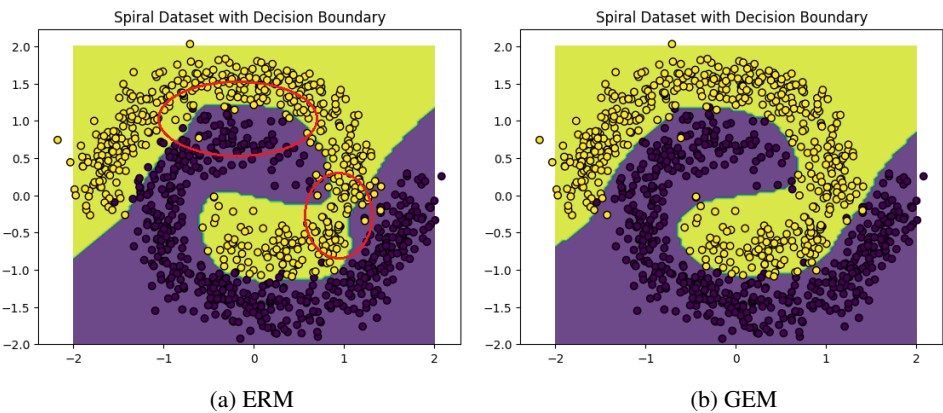

(a) ERM                                                      (b) GEM

Figure 6: Visualization of decision boundaries learned by a simple neural network using (a) ERM and (b) GEM on a synthetic spiral dataset.

## A.8    RESULTS ON TRANSFORMER-BASED MODELS

To verify the effectiveness of GEM on transformer-based models, we conduct more experiments on CIFAR-100 with 4 different transformer-based DNNs namely ViT-T (Xu et al., 2023), MobileViTv2-0.5 (Mehta & Rastegari, 2022), DeiT-Ti (Touvron et al., 2021), and EfficientFormer-L1 (Li et al., 2022). The training recipe is adopted from Xu et al. (2023) without modification and the hyperparameters of GEM remain the same as the ones reported in Section 5. Note that the training recipe for transformer-based models includes much stronger regularization such as mixup, cutmix, randaugment (Cubuk et al., 2020), random erasing (Zhong et al., 2020), label smoothing and so on. Results are presented in Table 6, demonstrating that GEM can achieve notable performance improvements over ERM, even in scenarios where strong regularization is already applied.

Table 6: Top-1 test accuracy (%) on CIFAR-100.

| Method | ViT-T | MobileViTv2-0.5 | DeiT-Ti | EfficientFormer-L1 |
|---|---|---|---|---|
| ERM | $71.84 \pm 0.08$ | $79.32 \pm 0.01$ | $71.79 \pm 0.11$ | $80.26 \pm 0.31$ |
| **GEM** | $\textbf{72.13} \pm 0.24$ | $\textbf{79.79} \pm 0.36$ | $\textbf{72.29} \pm 0.17$ | $\textbf{80.59} \pm 0.10$ |
| $\Delta$ | +0.29 | +0.48 | +0.51 | +0.33 |

## A.9    STANDARD DEVIATION FOR RESULTS IN FIGURES

For better reproducibility, we present in Fig. 7 the standard deviation bars for results in Figs. 2 and 3.

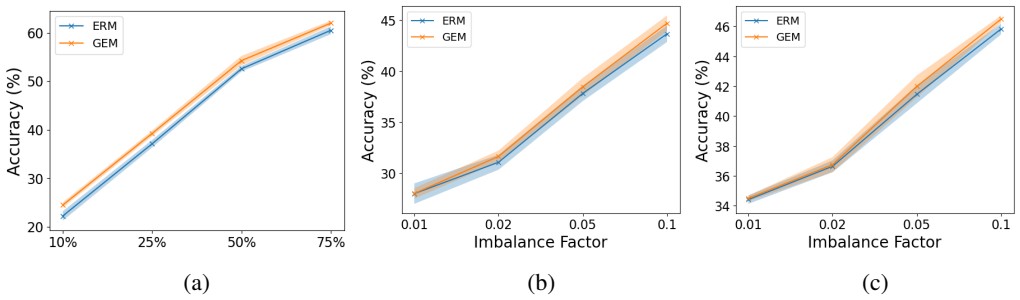

(a)                                      (b)                                      (c)

Figure 7: Standard deviation bars for results in (a) few-shot scenario, (b) long-tail imbalance scenario, and (c) step imbalance scenario.

### A.10 RESULTS ON FEW-SHOT TRANSFER LEARNING

In few-shot learning scenario, transfer learning often provides better performance than training from scratch. Therefore, instead of training a MobileNetV2 model on CIFAR-100 from scratch, we try to leverage the MobileNetV2 released by PyTorch which is pretrained on ImageNet. We completely freeze the feature extractor of the pretrained model and only fine-tune the classification head with 30 epochs in few-shot scenarios using different percentages of training samples. It turns out that transfer learning can indeed improve the performance at 10%, 25% and 50% configurations over training from scratch, while being worse than training from scratch at 75% scenario. Therefore, we compare between ERM and GEM only in 10%, 25% and 50% cases, as shown in Table 7. In general, GEM can still achieve some performance gain, especially when the data is more limited. However, the gains achieved by GEM are not as much as those it gets where the models are trained from scratch, since (1) the optimization is limited to the fully connected layer, and (2) not much overfitting is observed during fine-tuning.

Table 7: Top-1 test accuracy (%) on CIFAR-100.

|  | 10% | 25% | 50% |
|---|---|---|---|
| ERM | $47.36 \pm 0.26$ | $51.97 \pm 0.13$ | $55.27 \pm 0.21$ |
| **GEM** | $\mathbf{47.80} \pm 0.23$ | $\mathbf{52.11} \pm 0.19$ | $\mathbf{55.29} \pm 0.16$ |
| $\Delta$ | +0.45 | +0.14 | +0.02 |

