# OpenReview forum: "Generalization Error Minimized Deep Learning"
_ICLR.cc/2025/Conference — ICLR 2025 Conference Withdrawn Submission_

### Official Review · Reviewer_NT5P · 2024-11-01

**Soundness:** 1
**Presentation:** 2
**Contribution:** 1
**Rating:** 3
**Confidence:** 4

**Summary:**

This paper proposes a new algorithm called generalization error minimized deep learning (GEM DL). The algorithm builds upon a bias-variance decomposition for the generalization error. The decomposition then inspires a new form of regularization term to be added to the objective function. The authors empirically show models trained using the new algorithm experience better generalization performance compared to the naive empirical risk minimization.

**Strengths:**

- The paper studies a very interesting topic to the research community.
- Overall, the type of experimentations that are done to evaluate the effectiveness of the proposed objective function are good. There are however issues within the experiments which I’ll detail in the weaknesses section.

**Weaknesses:**

- Overall, the paper is difficult to read. There are various vague expressions/statements throughout the paper. Starting from the abstract, what is meant by “a new form of deep learning”? Or in the introduction “training/testing performance”? This ambiguity makes it difficult to follow the authors' approach, especially when specific details are introduced before fundamental concepts are defined. Further contributing to the confusion, Section 3.1 refers to "average loss" as "performance", which contradicts the conventional understanding of these terms.
- The motivation for introducing new definitions of generalization error and conditional generalization error remains unclear. The authors should clarify why they deviate from established definitions used in theoretical and empirical research. A detailed comparison between their proposed definitions (Equations 2 and 3) and conventional ones is necessary, highlighting the differences and justifications for these changes. Also, the definition in Equation 2 is more the “generalization gap” rather than the “generalization error”.
- Ultimately, the proposed optimization method reduces to adding a regularization term to the training loss (or empirical risk minimization objective function).  Therefore, comparisons with benchmark regularization algorithms are crucial. However, the empirical results are limited in this regard, with only one benchmark included. Moreover, this chosen competitor exhibits lower performance than baseline ERM on the ImageNet dataset (Table 2), suggesting it may not be a suitable benchmark for state-of-the-art comparisons.  Furthermore, most of the experiments omit this competitor entirely, presenting only comparisons to ERM, which is insufficient. Even if considered a "plug-and-play" method, the proposed algorithm should be compared against other techniques that incorporate regularization terms into the objective function.
- The application of the proposed method in Case 2 relies on a questionable assumption: knowledge of the distribution shift that will occur in the test case. In practice, both the type and amount of shift are often unknown a priori, raising concerns about the method's practicality in out-of-distribution scenarios. This limitation is evident in Figure 1, where the applied GEMs are pre-designed for the specific type of distribution shift. Moreover, when the amount of shift is unknown and an incorrect hyperparameter is estimated (e.g., overestimating the blurring amount with sigma=3 in Figure 1.b), performance deteriorates significantly compared to baseline ERM. This experiment highlights the method's sensitivity to accurate estimations of the distribution shift.


More minor weaknesses/comments:
- The statement "since  ˆθ depends only on D and the training process, it follows that T and ˆθ are independent" is inaccurate. Model parameters depend not only on the training data itself but also on the underlying training data distribution, which may not be independent of the test data distribution. The authors need to clarify the distinction between the independence of random sampling used to collect training and test data, and the potential dependence between the underlying distributions from which these samples are drawn.
- The proposed method is adding two more hyper-parameters to the learning method, which would require additional hyper-parameter selection, making the method less practically appealing.
- There are no standard deviation bars in any of the figures throughout the paper. This is particularly problematic in Figure 4 where there are lots of small variations.

**Questions:**

- In Section 3.1, the defined generalization error appears to be its out-of-distribution variant. The authors should clarify the rationale behind using this specific setting and explain why they refer to this generalization gap as a "new generalization error." What novelty does it introduce? Additionally, the presence of a squared term within the generalization error definition requires justification.

---

> ### Author Response · Authors · 2024-11-28
> **Response to Reviewer NT5P (Part 1)**
>
> We thank you very much for taking time to review our paper and provide valuable feedbacks. Below, please find our point-by-point responses to your comments.
>
> > Comment: Overall, the paper is difficult to read. There are various vague expressions/statements throughout the paper. Starting from the abstract, what is meant by “a new form of deep learning”? Or in the introduction “training/testing performance”? This ambiguity makes it difficult to follow the authors' approach, especially when specific details are introduced before fundamental concepts are defined. Further contributing to the confusion, Section 3.1 refers to "average loss" as "performance", which contradicts the conventional understanding of these terms.
>
> **Response**: Sorry for the confusion we might have caused. Let us clarify them here.
> - "a new form of deep learning" refers to our proposed generalization error minimized deep learning (GEM DL), where the generalization error is directly considered in the learning objective function. We believe the significant difference between GEM DL and the standard DL based on ERM suffices to support our claim that GEM DL is a new form of DL.
> - “training/testing performance” refers to the average training/testing loss as we defined in Section 3.1. We acknowledge that "risk" might be a more familiar terminology in generalization theory. However, the loss function of a learning task is often a close estimate of or exactly the same as the performance metric (NLL loss vs. 0-1 loss for classification and squared error for regression). Also, the average loss is often used to judge the performance in practice, e.g., determining the epoch for early stopping on the validation set. Therefore, we believe it's justified and acceptable to term "average loss" as "performance".
>
> > Comment: The motivation for introducing new definitions of generalization error and conditional generalization error remains unclear. The authors should clarify why they deviate from established definitions used in theoretical and empirical research. A detailed comparison between their proposed definitions (Equations 2 and 3) and conventional ones is necessary, highlighting the differences and justifications for these changes. Also, the definition in Equation 2 is more the “generalization gap” rather than the “generalization error”.
>
> **Response**: In the literature, there are many different ways to evaluate generalization, including “generalization gap” and “generalization error” as you mentioned. In our case, we interpret "generalization" as the difference between training and testing, which is intuitive and accepted by many in the literature. The difference between two objects, mathematically, is measured by distance. Therefore, the remaining problem is to determine which distance metric to use. In our case, we choose (the squared) $L^2$ distance since it's the popular Euclidean distance, and more importantly, it results in a nice bias-variance decomposition. Therefore, given that our new definitions of generalization error and conditional generalization error are both intuitive and useful, we believe it's justified to deviate from established definitions.

---

> ### Author Response · Authors · 2024-11-28
> **Response to Reviewer NT5P (Part 2)**
>
> > Comment: Ultimately, the proposed optimization method reduces to adding a regularization term to the training loss (or empirical risk minimization objective function). Therefore, comparisons with benchmark regularization algorithms are crucial. However, the empirical results are limited in this regard, with only one benchmark included. Moreover, this chosen competitor exhibits lower performance than baseline ERM on the ImageNet dataset (Table 2), suggesting it may not be a suitable benchmark for state-of-the-art comparisons. Furthermore, most of the experiments omit this competitor entirely, presenting only comparisons to ERM, which is insufficient. Even if considered a "plug-and-play" method, the proposed algorithm should be compared against other techniques that incorporate regularization terms into the objective function.
>
> **Response**: We sincerely appreciate your comments.
>
> One of the criticisms to deep learning from non-AI community is the lack of understanding and interpretability. If any modification to the optimization objective function in deep learning is brushed off as another regularization term, it will not bode well for dealing with that criticism. Yes, GEM eventually includes two extra terms in the optimization objective functions. However, those terms have their intended physical meaning, are related to generalization error, and offer some level of interpretability.
>
> As for more comparison, thank you for this suggestion. We now include dropout and a regularizer-based method (confidence penalty [1]) in Sec. A.5 of our revision as stronger benchmarks besides ERM and DOM. In general, GEM in Case 1 outperforms both of them by a clear margin. Please refer to Sec. A.5 for more details.
>
> |Method|MobileNetV2|ShuffleNetV2|WRN-40-2|resnet32|resnet20|vgg8|
> |:---:|:---:|:---:|:---:|:---:|:---:|:---:|
> |ERM|64.60|71.82|75.61|71.14|69.06|70.36|
> |CP|65.14 $\pm$ 0.77|**73.46** $\pm$ 0.25|76.35 $\pm$ 0.16|71.21 $\pm$ 0.25|69.07 $\pm$ 0.29|70.97 $\pm$ 0.25|
> |$\Delta$|+0.54|+1.64|+0.74|+0.07|+0.01|+0.61|
> |**GEM**|**65.99** $\pm$ 0.59|73.17 $\pm$ 0.30|**76.93** $\pm$ 0.31|**71.95** $\pm$ 0.30|**69.84** $\pm$ 0.18|**71.04** $\pm$ 0.18|
> |$\Delta$|+1.39|+1.35|+1.32|+0.81|+0.78|+0.68|
>
> |Method|WRN-40-2|WRN-40-1|WRN-16-2|
> |:---:|:---:|:---:|:---:|
> |ERM|75.61|71.98|73.26|
> |Dropout|76.68	$\pm$ 0.21|71.71 $\pm$ 0.19|73.33 $\pm$ 0.20|
> |$\Delta$|+1.07|-0.27|+0.07|
> |**GEM**|**76.93** $\pm$ 0.31|**72.09** $\pm$ 0.28|**73.61** $\pm$ 0.45|
> |$\Delta$|+1.32|+0.11|+0.35|
>
> Moreover, we also include a competitive benchmark, dubbed stability training (ST), to compare with GEM in Case 2. ST is specially designed to improve DNN’s generalization (robustness) to low quality images, whose loss is computed from both the training data $X$ and processed training data $\hat{X}$, thus providing a fair comparison with GEM in Case 2. Nonetheless, although ST is better than ERM in the presence of low quality images, GEM consistently outperforms it by a clear margin. Please refer to Sec. A.6 of our revision for more details.

---

> ### Author Response · Authors · 2024-11-28
> **Response to Reviewer NT5P (Part 3)**
>
> > Comment: The application of the proposed method in Case 2 relies on a questionable assumption: knowledge of the distribution shift that will occur in the test case. In practice, both the type and amount of shift are often unknown a priori, raising concerns about the method's practicality in out-of-distribution scenarios. This limitation is evident in Figure 1, where the applied GEMs are pre-designed for the specific type of distribution shift. Moreover, when the amount of shift is unknown and an incorrect hyperparameter is estimated (e.g., overestimating the blurring amount with sigma=3 in Figure 1.b), performance deteriorates significantly compared to baseline ERM. This experiment highlights the method's sensitivity to accurate estimations of the distribution shift.
>
> **Response**: We sincerely appreciate your thoughtful comment. However, the contributions of this paper should be evaluated from the following perspectives:
> - Generalization is a major theme in deep learning, with generalization error typically seen as a significant challenge in both theory and practice. This paper is the first to take a novel approach by transforming this challenge into an opportunity. We leverage generalization error in a constructive and positive manner to improve deep learning performance, supported by solid theoretical developments.
> - It is reasonable to assume that the distribution of testing data is related to the distribution of training data. Once a relationship between these distributions is established, it becomes theoretically possible to infer the distribution of testing data from that of training data. While the method for such inference is a complex question that warrants further investigation, it falls beyond the scope of this paper. What this paper demonstrates is that, if the distribution of testing data can be inferred from the distribution of training data, then GEM can be applied to leverage generalization error to enhance deep learning performance.
> - Case 2 is just one example we selected to illustrate the above concepts. We do not claim that Case 2 is representative of all OOD (out-of-distribution) application scenarios. However, it is a practical and relevant example that helps clarify our approach.
>
> As for Figure 1, we argue that GEM in Case 2 is not sensitive to the estimated amount of distribution shift. In Figure 1(b), when we overestimate the blurring amount with $\sigma=3$, our performance is still better than ERM at $\sigma=1$; when we underestimate the blurring amount with $\sigma=1$, our performance is also better than ERM at $\sigma=3$. Similarly, in Figure 1(a), when we overestimate the compression amount with $q=10$, our performance is still better than ERM at $q=30$; when we underestimate the compression amount with $q=30$, our performance is also better than ERM at $q=10$. In fact, in both JPEG compression and Gaussian bluring scenarios, no matter how we estimate the distribution shift, the resulting GEM is better than ERM in most distortion levels.

---

> ### Author Response · Authors · 2024-11-28
> **Response to Reviewer NT5P (Part 4)**
>
> > Comment: The statement "since $\hat{θ}$ depends only on D and the training process, it follows that T and $\hat{θ}$ are independent" is inaccurate. Model parameters depend not only on the training data itself but also on the underlying training data distribution, which may not be independent of the test data distribution. The authors need to clarify the distinction between the independence of random sampling used to collect training and test data, and the potential dependence between the underlying distributions from which these samples are drawn.
>
> **Response**: Sorry, you seem to confuse the meaning of independence in probability theory with the literal meaning of independence in plain English language.
>
> To avoid confusion in discussion, we'd like to first clarify the word "depend".
> - In the context of "since $\hat{θ}$ depends only on D and the training process, it follows that T and $\hat{θ}$ are independent", "depend/independent" is a terminology  in probability theory, i.e., $\hat{θ}$ and D are dependent random variables, while $\hat{θ}$ and T are independent random variables.
> - In the context of “Model parameters depend … on the underlying training data distribution, which may not be independent of the test data distribution.”, “depend/independent” takes its literal meaning in plain English language, since “dependence/independence” in probability theory is only defined between events, classes of events, or random variables, but not between distributions (which are measures). Therefore, to avoid confusion, let’s use “rely/irrelevant” instead of “depend/independent” in this context.
>
> With the above clarification, we can now address your confusion.
> - The statement "since $\hat{θ}$ depends only on D and the training process, it follows that T and $\hat{θ}$ are independent" is indeed accurate in the sense of probability theory, allowing us to conduct all the proposed mathematical derivations. It is common knowledge that the testing data $T$, which is regarded as random samples, must not and should not be used during the training phase. Since $D$ and $T$ are independent in the sense of probability theory, so are $T$ and $\hat{\theta}$ in the sense of probability theory.
> - Your comment “Model parameters **rely**… on the underlying training data distribution, which may not be **irrelevant** of the test data distribution.” is also accurate. In fact, the relationship between the test data distribution and the training data distribution is pivotal in our generalization proxies. As we mentioned in Line 246~247 of our submission, the proposed generalization proxy does **not** depend on the test data itself; instead, it only relies on **the distribution of the test data**. Therefore, we consider two cases, where in Case 1 the distribution of the test data is **identical** to that of the training data (which is typically assumed in most learning tasks), or in Case 2 the distribution of the test data **can be inferred from** the training data. In Case 1, we can replace the expectation over the pair of random variables $(U, V)$ by the expectation over the pair of random variables $(X, Y)$ since these two quantities are equal given the distributional equivalence. Similarly, we replace the expectation over the pair of random variables $(U, V)$ by the expectation over the pair of random variables $(\hat{X}, Y)$ in Case 2, again, leveraging the distributional equivalence.
>
> > Comment: The proposed method is adding two more hyper-parameters to the learning method, which would require additional hyper-parameter selection, making the method less practically appealing.
>
> **Response**: We've made a discussion about this in the Appendix of our original submission. Please refer to Sec. A.3 for more details. In fact, GEM is generally not sensitive to hyperparameters tuning. For example, for all additional experiments we conducted for GEM during rebuttal (using new CNN and tranformer models, or doing transfer learning instead of training from scratch), we have directly reused the same hyperparameters as the ones we reported in our original submission without any fine-tuning. Even with different models and different training settings, the previous hyperparameters are still vaild and can maintain the effectiveness of GEM.

---

> ### Author Response · Authors · 2024-11-28
> **Response to Reviewer NT5P (Part 5)**
>
> > Comment: There are no standard deviation bars in any of the figures throughout the paper. This is particularly problematic in Figure 4 where there are lots of small variations.
>
> **Response**: Thank you for this suggestion. In response, we have included the standard deviation bars for Figure 2 & 3 in Sec. A.9 of our revision. However, Figure 4 is not intended to report performance results but rather to analyze and visualize the learning process. Although in the early stage of learning, there are a lot of fluctuations in curves (which is natural, as the initial learning rate is high), only the last stage is of particular interest, since it corresponds to the final trained model. As clearly illustrated in Figure 4, there's no much fluctuation in curves at the end of training. Combined with the fact that the gaps between GEM and ERM are significant, we believe it is unnecessary to include standard deviation bars in Figure 4. Similarly, standard deviation bars were not included for Figure 1, as it is conventionally uncommon to conduct multiple runs on ImageNet, where results typically exhibit minimal variation.
>
> > Comment: In Section 3.1, the defined generalization error appears to be its out-of-distribution variant. The authors should clarify the rationale behind using this specific setting and explain why they refer to this generalization gap as a "new generalization error." What novelty does it introduce? Additionally, the presence of a squared term within the generalization error definition requires justification.
>
> **Response**: To the best of our knowledge, we couldn't find any other papers in the literature using the same form of (conditional) generalization error as us. We'd appreciate it if you could point us to some literature where the same definition is adopted so that we can refer to them. As for the justification of our generalization error definition, please refer to our response to your second comment.
>
> [1] Gabriel Pereyra, George Tucker, Jan Chorowski, Łukasz Kaiser, and Geoffrey Hinton. Regularizing neural networks by penalizing confident output distributions. arXiv preprint arXiv:1701.06548, 2017. \
> [2] Stephan Zheng, Yang Song, Thomas Leung, and Ian Goodfellow. Improving the robustness of deep neural networks via stability training. CVPR 2016.
>
> **We thank you again for your time and effort to review our paper.  If our responses above clear your misunderstandings and address all your concerns, could you kindly increase your score to a higher level.**

---

> ### Author Response · Authors · 2024-12-02
>
> Dear Reviewer NT5P,
>
> Thanks again for reviewing our paper. Since the discussion period is going to end soon, we are eager to know if you are satisfied with our previous response. If no, please kindly tell us your remaining concerns and hopefully we can address them before the deadline. If yes, we wonder if it’s possible for you to raise the score. In any case, we would be extremely grateful to hear from you.
>
> Thanks.

---

### Official Review · Reviewer_fhoH · 2024-11-02

**Soundness:** 3
**Presentation:** 2
**Contribution:** 3
**Rating:** 5
**Confidence:** 2

**Summary:**

This paper introduces generalization error minimized (GEM) deep learning, a method that optimizes the standard minimization objective coupled with a “generalization” proxy target. The authors first decompose the conditional generalization error into three terms: Conditional testing variance, Conditional training variance and bias between the expected training and testing performance. The authors then discuss two cases: Case 1 when the training and test distribution are similar and Case 2 where the train and test distribution are dissimilar. In case 1, the GEM objective boils down to the linear combination of cross entropy (CE) loss, square of the CE loss and the second moment of the CE loss. Results with GEM in image classification task on CIFAR, ImageNet with ResNet, MobileNetV2 and other backbones shows that GEM performs better than ERM.

**Strengths:**

The authors show experimental results on a wide range of settings include few-shot setup, imbalanced learning in the context of image classification. The results on various datasets and settings show that GEM performs better than ERM in most of the scenarios.

**Weaknesses:**

1. I am not convinced the by the Case 2 argument. In most cases, the distribution (U, V) can not be obtained by processing (X, Y). For example, consider the case of subgroup shift (Waterbirds, Celeb-A) or OOD datasets (ImageNet-Sketch, ImageNet-R etc).
2. The authors should also report performance on various OOD datasets atleast in the case of ImageNet (for e.g ImageNet-A, ImageNet-Sketch to name a few)
3. It would be great if the authors can also show a toy experiment with a synthetic data and how the solution learned by GEM and ERM differs

**Questions:**

1. Is label smoothing used in both ERM and GEM? If yes, what is the alpha value? The pseudocode in Appendix A.2 mentions label smoothing.
2. In case of Case 2, the authors should compare against a method that computes CE loss w.r.t $\hat{X}$ and original CE loss and not the general ERM.
3. Could the authors provide the detailed hyperparameter and regularization configurations in each of the setups. In L524-529, the authors mention that many regularization techniques are already included in the baseline method. I could not find the relevant details anywhere in the paper.

---

> ### Author Response · Authors · 2024-11-28
> **Response to Reviewer fhoH (Part 1)**
>
> We thank you very much for taking time to review our paper and provide valuable feedbacks. Below, please find our point-by-point responses to your comments.
>
> > Comment: I am not convinced the by the Case 2 argument. In most cases, the distribution (U, V) can not be obtained by processing (X, Y). For example, consider the case of subgroup shift (Waterbirds, Celeb-A) or OOD datasets (ImageNet-Sketch, ImageNet-R etc).
>
> **Response**: We sincerely appreciate your thoughtful comment. However, the contributions of this paper should be evaluated from the following perspectives:
> - Generalization is a major theme in deep learning, with generalization error typically seen as a significant challenge in both theory and practice. This paper is the first to take a novel approach by transforming this challenge into an opportunity. We leverage generalization error in a constructive and positive manner to improve deep learning performance, supported by solid theoretical developments.
> - It is reasonable to assume that the distribution of testing data is related to the distribution of training data. Once a relationship between these distributions is established, it becomes theoretically possible to infer the distribution of testing data from that of training data. While the method for such inference is a complex question that warrants further investigation, it falls beyond the scope of this paper. What this paper demonstrates is that, if the distribution of testing data can be inferred from the distribution of training data, then GEM can be applied to leverage generalization error to enhance deep learning performance.
> - Case 2 is just one example we selected to illustrate the above concepts. We do not claim that Case 2 is representative of all OOD (out-of-distribution) application scenarios. However, it is a practical and relevant example that helps clarify our approach.
>
> > Comment: The authors should also report performance on various OOD datasets atleast in the case of ImageNet (for e.g ImageNet-A, ImageNet-Sketch to name a few).
>
> **Response**: Please see our response to the previous comment. General OOD generalization is certainly an interesting topic, but beyond the scope of this paper. Addressing it would divert our attention away from our purpose, and won’t support our main contributions.
>
> > Comment: It would be great if the authors can also show a toy experiment with a synthetic data and how the solution learned by GEM and ERM differs.
>
> **Response**: Thanks for this insightful comment. We have included a toy experiment on the classical spiral dataset in Sec. A.7 of our revision. As elaborated therein, the model learned by GEM not only has higher testing accuracy than the one learned by ERM, but also has smoother decision boundary, showing that it successfully captures the true feature of the data instead of overfitting the noise. Please refer to Sec. A.7 for more details and the decision boundary visualization.
>
> > Comment: Is label smoothing used in both ERM and GEM? If yes, what is the alpha value? The pseudocode in Appendix A.2 mentions label smoothing.
>
> **Response**: Yes, for some models on some datasets. We will provide detailed explanation in the response to your last comment.
>
> > Comment: In case of Case 2, the authors should compare against a method that computes CE loss w.r.t  $\hat{X}$  and original CE loss and not the general ERM.
>
> **Response**: Thank you for pointing it out. To echo this, we have added a comparsion between GEM and stability training (ST) [1] in Sec. A.6 of our revision. ST is specially designed to improve DNN’s generalization (robustness) to low quality images, whose loss is computed from both the training data $X$ and processed training data $\hat{X}$, thus providing a fair comparison with GEM in Case 2. Nonetheless, although ST is better than ERM in the presence of low quality images, GEM consistently outperforms it by a clear margin. Please refer to Sec. A.6 for more details.

---

> ### Author Response · Authors · 2024-11-28
> **Response to Reviewer fhoH (Part 2)**
>
> > Comment: Could the authors provide the detailed hyperparameter and regularization configurations in each of the setups. In L524-529, the authors mention that many regularization techniques are already included in the baseline method. I could not find the relevant details anywhere in the paper.
>
> **Response**: The configuration of our experiments on CIFAR-100 can be found in [2] (or directly in the code we provided in our supplementary materials), while the recipes on ImageNet can be found on the PyTorch official website (https://github.com/pytorch/vision/tree/main/references/classification), both mentioned in Sec. 5.1 of our original submission. Moreover, we have presented additional experiments for transformer-based models on CIFAR-100 in Sec. A.8 as requested by another reviewer, whose training recipe can be found in Appendix A of [3]. Overall, the regularization techniques in those training recipes include weight decay, regular data augmentation (random crop and flip), mixup, cutmix, randaugment [4], label smoothing (LS) and so on. In terms of LS, $\alpha=0.1$ for both ShuffleNetV2 on ImageNet and transformer-based models on CIFAR-100, while other experiments don’t include LS.
>
> [1] Stephan Zheng, Yang Song, Thomas Leung, and Ian Goodfellow. Improving the robustness of deep neural networks via stability training. CVPR 2016. \
> [2] Yonglong Tian, Dilip Krishnan, and Phillip Isola. Contrastive representation distillation. ICLR 2019. \
> [3] Zhiqiu Xu, Yanjie Chen, Kirill Vishniakov, Yida Yin, Zhiqiang Shen, Trevor Darrell, Lingjie Liu, and Zhuang Liu. Initializing models with larger ones. ICLR 2024. \
> [4] Ekin D Cubuk, Barret Zoph, Jonathon Shlens, and Quoc V Le. Randaugment: Practical automated data augmentation with a reduced search space. CVPRW 2020.
>
> **We thank you again for your time and effort to review our paper.  If our responses above address all your concerns, could you kindly increase your score to a higher level.**

---

> ### Author Response · Authors · 2024-12-02
>
> Dear Reviewer fhoH,
>
> Thanks again for reviewing our paper. Since the discussion period is going to end soon, we are eager to know if you are satisfied with our previous response. If no, please kindly tell us your remaining concerns and hopefully we can address them before the deadline. If yes, we wonder if it’s possible for you to raise the score. In any case, we would be extremely grateful to hear from you.
>
> Thanks.

---

> > ### Comment · Reviewer_fhoH · 2024-12-02
> > **Response to Rebuttal**
> >
> > I thank the authors for the detailed rebuttal.
> > - The synthetic toy experiment in A.7 is definitely interesting.
> > - I thank the authors for clarifying the experimental details.
> > - The additional comparison against Stability training is great as well.
> > - My biggest concern still remains Case 2 argument. The authors make a case that "if the distribution of testing data can be inferred from the distribution of training data, then GEM can be applied". I am not still convinced by the argument that it is tractable to infer the distribution of test data.

---

> > > ### Author Response · Authors · 2024-12-02
> > >
> > > We thank you very much for your appreciation of our rebuttal effort. Let us further clarify our view on the Case 2 argument.
> > >
> > > In fact, the Case 1 scenario is just a special case of the Case 2 scenario. Assuming the distribution of testing data $(U, V)$ to be identical to that of the training data $(X, Y)$ is a living example of **inferring the distribution of testing data from the distribution of training data**. This inference is indeed tractable and used universally in machine learning implicitly.
> > >
> > > In our Case 2 application, we simply extend the above inference one step further by assuming that $(U, V)$ has the same distribution as the processed training data $(\hat{X}, Y)$, which is also tractable.
> > >
> > > Again, we thank you for your reviewing effort. If our response above address your concerns, could you kindly increase your score to a higher level.

---

### Official Review · Reviewer_i3N6 · 2024-11-03

**Soundness:** 1
**Presentation:** 3
**Contribution:** 2
**Rating:** 3
**Confidence:** 4

**Summary:**

This paper introduces a framework that integrates a 'novel' bias-variance decomposition of generalization error into the training objective for deep neural networks (DNNs). The proposed framework aims to improve the model's generalization by jointly optimizing the traditional ERM loss and a theoretically-inspired proxy for generalization error, yielding gains in accuracy  across various image classification scenarios.

**Strengths:**

1- The paper is extremely well-written and structured.
2-  Generalization in deep learning is a very important problem. The proposed generalization decomposition is simple, yet sound and interesting.

**Weaknesses:**

W1- From a theoretical perspective, the generalization decomposition is quite straight-forward for me. So the main contribution here lies in leveraging the theoretical bound to propose a practical proxy of generalization. Regarding this proposed proxy, I have some concerns (See the question bellow) regarding the link between the theory and practice.

W2- The experiments focused only on image classification tasks with CNN-based structure with NLL loss. I would like to have seen a more diversified  experiments, e.g.,  transformer-based models. regression tasks, NLP, to better evaluate the utility of the proposed approach.

I will reconsider my score if the authors can address my concerns in the questions section.

**Questions:**

**Theory**

In eq 11, the first two terms use the test data (True population). The third term is based on the training data and the forth uses both distribution. In the remaining part of the paper (after eq 11 Section 4.1), the authors argue that we can neglect the last two terms and keep only the first 2 (defined using the population data T ). However, in the experiment section, the proposed generalization proxy is based on the training data D and not on independent data). This poses several contradictions:

Q-1: The proposed proxy does not minimize  $\text{Var}(\Omega(T, \hat{\theta}) | \hat{\theta}) + K^2(\hat{\theta})$. In fact, as it is based on the training data, what it really minimizes is $\text{Var}(\Omega(D, ) | \hat{\theta}) + KJ^2(\hat{\theta})$, which contradicts the argument made by the author (between eq11 and 13) that these two terms should be ignored.

Q-2:  With the proposed proxy setup, T and $\hat{\theta}$ are no longer independent. In fact $\hat{\theta}$ is optimized using T. Hence, Eq 4 and onwards are no longer valid.

Q-3: One potential way to solve this in my opinion is to define the proxy using an auxiliary validation set, i.e., in equation 19, the first term is defined using the training data while the second and third are computed using an independent validation data.


**Experiments**

Q-4: Even that the link between the theory and the generalization proxy is not valid or weak (Q1-3 above), I think that the proposed regularizer can still be useful. However. the empirical validation in this paper is not enough to evaluate that. Have the authors tried their regularizer with other architectures beside CNN and NLL loss, e.g., transformer?

Q-5. Empirically, the authors compare only with DOM. However, several other theoretically-sound approaches have been proposed in the literature that can reduce overfitting, e.g., dropout [1-2], Mixup [3-4], feature diversity [5-7], etc.


[1] Srivastava, Nitish, et al. "Dropout: a simple way to prevent neural networks from overfitting." The journal of machine learning research 15.1 (2014): 1929-1958. \
[2] Wan, Li, et al. "Regularization of neural networks using dropconnect." International conference on machine learning. PMLR, 2013.  \
[3] Zhang, Hongyi. "mixup: Beyond empirical risk minimization." arXiv preprint arXiv:1710.09412 (2017). \
[4] Zhang, Linjun, et al. "How does mixup help with robustness and generalization?."  Neurips (2020). \
[5] Laakom, Firas, et al. "WLD-reg: A data-dependent within-layer diversity regularizer." Proceedings of the AAAI Conference on Artificial Intelligence. Vol. 37. No. 7. 2023. \
[6] Laakom, Firas, et al. "Learning distinct features helps, provably." ECML, 2023.
[7] Cogswell, Michael, et al. "Reducing overfitting in deep networks by decorrelating representations." ICLR 2016

---

> ### Author Response · Authors · 2024-11-28
> **Response to Reviewer i3N6 (Part 1)**
>
> We thank you very much for taking time to review our paper and provide valuable feedbacks. Below, please find our point-by-point responses to your comments.
>
> > Comment: In eq 11, the first two terms use the test data (True population). The third term is based on the training data and the forth uses both distribution. In the remaining part of the paper (after eq 11 Section 4.1), the authors argue that we can neglect the last two terms and keep only the first 2 (defined using the population data T ). However, in the experiment section, the proposed generalization proxy is based on the training data D and not on independent data.
>
> **Response**: Our theoretic derivation is solid. Our theory and experiments are consistent. There is no contradiction at all.
>
> To help you understand our paper better, we would like to point out an important conceptual distinction between testing data and its probability distribution. While these two concepts are related, they are fundamentally different. Testing data refers to specific data samples (i.e., datasets) used exclusively for evaluation, which must not and should not be used during the training phase. In contrast, the probability distribution of testing data represents the general probabilistic characteristics of the data population, which is typically assumed to be related to, equal to, or inferred from, the probability distribution of the training data. Confusing these concepts can lead to significant misunderstandings and make it difficult to fully grasp the content of our paper. We kindly ask you to keep this distinction in mind as you review our responses to your comments below.
> - Firstly, we would like to clarify a misunderstanding in the comment “In eq 11, the first two terms use the test data (True population)”. No terms in Eq. (11) use test data, which, as mentioned above, must not and should not be used during the training phase. All quantities in Eq. (11) depend on the respective distributions, but not the data itself. In particular, the first two terms in Eq. (11) depend on the distribution of testing data, but not the testing data itself.
> - Secondly, the shift from the distribution of testing data to the distribution of training data was clearly explained in Line 246\~269 of our original submission. Let us reiterate and clarify our reasoning here. As we mentioned in Line 246\~247 of our original submission, the proposed generalization proxy does **not** rely on the test data itself; instead, it only relies on **the distribution of the test data**. Therefore, we consider two cases, where in Case 1 the distribution of the test data is identical to that of the training data (which is typically assumed in most learning tasks), or in Case 2 the distribution of the test data can be inferred from the training data. In Case 1, we can replace the expectation over the pair of random variables $(U, V)$ by the expectation over the pair of random variables $(X, Y)$ since these two quantities are equal given the distributional equivalence. Similarly, we replace the expectation over the pair of random variables $(U, V)$ by the expectation over the pair of random variables $(\hat{X}, Y)$ in Case 2, again, leveraging the distributional equivalence.
>
> > Comment: The proposed proxy does not minimize $\text{Var}(\Omega(T,\hat{\theta})|\hat{\theta})+K^2(\hat{\theta})$. In fact, as it is based on the training data, what it really minimizes is $\text{Var}(\Omega(D,\hat{\theta})|\hat{\theta})+J^2(\hat{\theta})$, which contradicts the argument made by the author (between eq11 and 13) that these two terms should be ignored.
>
> **Response**: Please refer to our reply to the previous comment. Here you seem to have two confusions. The first confusion is about the testing data and its distribution. Given $\theta$, $\text{Var}(\Omega(T,\hat{\theta})|\hat{\theta})+K^2(\hat{\theta})$ depends on the distribution of the testing data only, but not the testing data itself. The second confusion is about $\text{Var}(\Omega(D,\hat{\theta})|\hat{\theta})+J^2(\hat{\theta})$. The quantity $\text{Var}(\Omega(D,\hat{\theta})|\hat{\theta})+J^2(\hat{\theta})$ is a different animal, which is totally different from $\text{Var}(\Omega(T,\hat{\theta})|\hat{\theta})+K^2(\hat{\theta})$ even when the testing data $T$ and the training data $D$ have the same distribution. Why? There are two reasons: (1) $T$ and $\hat{\theta}$ are independent, and hence $\text{Var}(\Omega(T,\hat{\theta})|\hat{\theta})+K^2(\hat{\theta})$ can be computed analytically, as shown in the derivation of Eq. (15); and (2) $\hat{\theta}$ depends on $D$, and $\text{Var}(\Omega(D,\hat{\theta})|\hat{\theta})+J^2(\hat{\theta})$ is highly convoluted to the extent that there is no formula for computing it, which is why we developed Proposition 1 to establish an upper bound.
>
> Please see our derivations in Line 246~269 of our original submission. What we minimize is exactly $\text{Var}(\Omega(T,\hat{\theta})|\hat{\theta})+K^2(\hat{\theta})$.

---

> ### Author Response · Authors · 2024-11-28
> **Response to Reviewer i3N6 (Part 2)**
>
> > Comment: With the proposed proxy setup, $T$ and $\hat{θ}$ are no longer independent. In fact $\hat{θ}$ is optimized using $T$. Hence, Eq 4 and onwards are no longer valid.
>
> **Response**: This comment is incorrect. Please see our replies to previous comments. It is common knowledge that the testing data $T$ must not and should not be used during the training phase. $\hat{θ}$ depends on the training data $D$ only. Since $D$ and $T$ are independent, so are $T$ and $\hat{θ}$. All derivations in the paper are mathematically solid.
>
> > Comment: One potential way to solve this in my opinion is to define the proxy using an auxiliary validation set, i.e., in equation 19, the first term is defined using the training data while the second and third are computed using an independent validation data.
>
> **Response**: Thanks for your suggestion. However, since our theory and generalization proxy are correct and self-consistent, there’s no need to involve any hold-out validation set.
>
> > Comment: Have the authors tried their regularizer with other architectures beside CNN and NLL loss, e.g., transformer?
>
> **Response**: Thank you for pointing it out. To echo this, we have included more experimental results for 4 transfomer-based models in Sec. A.8 of our revision. Although the standard training recipe of those models have already included a lot of regularization techiques, GEM can still further improve the generalization performance of them significantly. Please refer to Sec. A.8 for more details.
>
> |Method|ViT-T|MobileViTv2-0.5|DeiT-Ti|EfficientFormer-L1|
> |:---:|:---:|:---:|:---:|:---:|
> |ERM|71.84 $\pm$ 0.08|79.32 $\pm$ 0.01|71.79 $\pm$ 0.11|80.26 $\pm$ 0.31|
> |**GEM**|**72.13** $\pm$ 0.24|**79.79** $\pm$ 0.36|**72.29** $\pm$ 0.17|**80.59** $\pm$ 0.10|
> |$\Delta$|+0.29|+0.48|+0.51|+0.33|
>
> > Comment: Empirically, the authors compare only with DOM. However, several other theoretically-sound approaches have been proposed in the literature that can reduce overfitting, e.g., dropout, Mixup, feature diversity, etc.
>
> **Response**: Thank you for this suggestion. Regarding more comparison, we include dropout and a regularizer-based method (confidence penalty [1]), as suggested by reviewer NT5P, in Sec. A.5 of our revision as stronger benchmarks besides ERM and DOM. In general, GEM outperforms both of them by a clear margin. Please refer to Sec. A.5 for more details.
>
> |Method|MobileNetV2|ShuffleNetV2|WRN-40-2|resnet32|resnet20|vgg8|
> |:---:|:---:|:---:|:---:|:---:|:---:|:---:|
> |ERM|64.60|71.82|75.61|71.14|69.06|70.36|
> |CP|65.14 $\pm$ 0.77|**73.46** $\pm$ 0.25|76.35 $\pm$ 0.16|71.21 $\pm$ 0.25|69.07 $\pm$ 0.29|70.97 $\pm$ 0.25|
> |$\Delta$|+0.54|+1.64|+0.74|+0.07|+0.01|+0.61|
> |**GEM**|**65.99** $\pm$ 0.59|73.17 $\pm$ 0.30|**76.93** $\pm$ 0.31|**71.95** $\pm$ 0.30|**69.84** $\pm$ 0.18|**71.04** $\pm$ 0.18|
> |$\Delta$|+1.39|+1.35|+1.32|+0.81|+0.78|+0.68|
>
> |Method|WRN-40-2|WRN-40-1|WRN-16-2|
> |:---:|:---:|:---:|:---:|
> |ERM|75.61|71.98|73.26|
> |Dropout|76.68	$\pm$ 0.21|71.71 $\pm$ 0.19|73.33 $\pm$ 0.20|
> |$\Delta$|+1.07|-0.27|+0.07|
> |**GEM**|**76.93** $\pm$ 0.31|**72.09** $\pm$ 0.28|**73.61** $\pm$ 0.45|
> |$\Delta$|+1.32|+0.11|+0.35|
>
> Moreover, we’d like to emphasize that our proposed method is orthogonal to other regularization techiques. For example, although Mixup is used in the training of ShuffleNetV2 on ImageNet (https://github.com/pytorch/vision/tree/main/references/classification#shufflenet-v2) and those transformer-based models we provide in the revision, yet we can  further improve their performance.
>
> [1] Gabriel Pereyra, George Tucker, Jan Chorowski, Łukasz Kaiser, and Geoffrey Hinton. Regularizing neural networks by penalizing confident output distributions. arXiv preprint arXiv:1701.06548, 2017.
>
> **We thank you again for your time and effort to review our paper.  If our responses above clear your misunderstandings and address all your concerns, could you kindly increase your score to a higher level.**

---

> ### Author Response · Authors · 2024-12-02
>
> Dear Reviewer i3N6,
>
> Thanks again for reviewing our paper. Since the discussion period is going to end soon, we are eager to know if you are satisfied with our previous response. If no, please kindly tell us your remaining concerns and hopefully we can address them before the deadline. If yes, we wonder if it’s possible for you to raise the score. In any case, we would be extremely grateful to hear from you.
>
> Thanks.

---

> ### Comment · Reviewer_i3N6 · 2024-12-02
>
> I thank the reviewers for their response. While i appreciate the additional experiments, my concerns about the theory still stand and the authors have failed to address them: 'the theory itself is self-consistent. However, the proposed proxy is not consistent with the theory'.
>
> As the authors emphasized this is mainly a theoretical paper. However, as mentioned in the original review from a theoretical perspective, the 'proposed' generalization decomposition is quite straight-forward for me (e.g., no novel proof technique). So the main contribution here lies in leveraging the theoretical bound to propose a practical proxy of generalization.
>
> First, I would like to point out to, unlike what the authors are suggesting, this not the case of a confusion between the data and its distribution.  Indeed in eq11, the first 2 terms are defined using the population data T= {= {(u1, v1),(u2, v2), . . . ,(um, vm)}} (the test set as mentioned in L146). The other 2 last terms are defined using D = {(x1, y1),(x2, y2), . . . ,(xn, yn)} the training set. Additionally, the proxy you proposed is based on the data and not the probability distribution itself.  The proposed proxy minimizes $\text{Var}(\Omega(D, ) | \hat{\theta}) + KJ^2(\hat{\theta})$ and not $\text{Var}(\Omega(T, \hat{\theta}) | \hat{\theta}) + K^2(\hat{\theta})$. This yields a contradiction to the authors' argument (between eq11 and 13) that these two terms should be ignored. Furthermore, with the proposed proxy setup, T and $\hat{\theta}$ are no longer independent. In fact $\hat{\theta}$ is optimized using T.

---

> ### Author Response · Authors · 2024-12-02
>
> In our context, we regard the generalization error decomposition and the generalization proxies as a whole and refer to them as our "theory". We firmly believe that our theory, both the decomposition part and the proxies part, is solid.
>
> As for your concerns, we have already addressed them in detail in our previous response. But let us further explain our reasoning here.
>
> - The conditional generalization error is decomposed to four terms as shown in Eq. (11).
> - Following our argument in Line 217~237, we show that the last two terms are negligible, thus reaching the proxy in Eq. (13) by ignoring the last two terms in Eq. (11).
> - In Eq. (15),  we further derive that the proxy in Eq. (13) can be expressed by the combination of two expectations over the the pair of random variables $(U, V)$. **Note that the expectation only relies on the probability distribution of $(U, V)$ but not any specific data samples.** So the comment "the proxy you proposed is based on the data and not the probability distribution itself" is incorrect.
> - Finally, in Eq. (16) and (17), we replace the expectation over $(U, V)$ by the expectation over $(X, Y)$ in Case 1, where we assume the testing data has the same distribution as the training data (**if the distributions are identical, the expectations are equal**); and we replace the expectation over $(U, V)$ by the expectation over $(\hat{X}, Y)$ in Case 2, where we assume the testing data has the same distribution as the processed training data $(\hat{X}, Y)$.
>
> Moreover, our derivations are based on the fact that $T$ and $\hat{θ}$ are independent, which is indeed the case. $T$, as testing data samples, is never used during training, which is a common sense in the machine learning community. Therefore, the optimized model weights $\hat{θ}$ has no chance to be dependent on $T$.
>
> To further clarify our point, let's use the vanilla ERM as an example. When we train a DNN, what we really hope to minimize is the expected testing performance (population risk) $\mathbb{E}[\mathcal{L}(f_{\theta}(U), V)]$. However, we cannot and should not get access to the testing data samples during training, so we replace the expectation over $(U, V)$ by the expectation over $(X, Y)$, where we assume the testing data has the same distribution as the training data, thus resulting in the canonical ERM $\min_{\theta} \mathbb{E} [\mathcal{L}(f_{\theta}(X), Y)]$. Although $(X, Y)$ and $(U, V)$ are identically distributed, they are also independent. Therefore, the trained model weights $\hat{\theta}$ based on $D = \\{(x_1, y_1), (x_2, y_2), \dots, (x_n, y_n)\\}$, is naturally independent of $T = \\{(u_1, v_1), (u_2, v_2), \dots, (u_m, v_m)\\}$.
>
> **As you can see in the above example, all assumptions used in GEM are also used in ERM. As long as ERM is theoretically sound, so is GEM.**
>
> Again, we thank you for your reviewing effort. If our response above address your concerns, could you kindly increase your score to a higher level.

---

> ### Author Response · Authors · 2024-12-02
>
> To further clarify our point, we'll present another simple example here.
>
> Let $A=\\{Z_1, Z_2, \dots, Z_n\\}$ be a sequence of i.i.d. random variables with $Z_1 \sim \mathbb{P}$. Then, the value of $\mathrm{Var}(Z_1+Z_2+\dots+Z_n)=n\mathrm{Var}(Z_1)$ depends only on $\mathbb{P}$, but not $A$.

---

### Official Review · Reviewer_41QK · 2024-11-05

**Soundness:** 2
**Presentation:** 2
**Contribution:** 2
**Rating:** 5
**Confidence:** 3

**Summary:**

This paper introduces a new algorithm of DNN training named generalization error minimized (GEM). It is motivated by theoretical analyses of reducing the generalization error. Targeting on minimizing a squared generalization error, the authors derive that the squared expected test loss and expected squared test loss are the two key components for reducing generalization error. GEM is designed based on this theoretical result. Experiments CIFAR-100 and ImageNet confirm the effectiveness of GEM against ERM and DOM. Additional experiments of test data shift, few-shot learning and imbalanced classification also show GEM surpasses ERM.

**Strengths:**

1. This paper proposes a new algorithm of DNN training derived from generalization error minimization.
2. The algorithm is easy to implement and doesn’t require heavy computations.
3. GEM shows clear benefits in small datasets.

**Weaknesses:**

1. The theoretical part is a bit tricky to me. Eventually the generalization error is reduced to be two loss terms only relying on test data as shown in Eq (15). Then the algorithm is implemented by simply replacing the test data with training data or processed training data. For the latter case, the difference between training and test data must be known in advance, which is not very realistic. Also, the theoretical result is based on the particular form of squared generalization error, which is different from the common definition of generalization error (training loss - test loss). The motivation of using the squared form of generalization error should be explained.
2. The optimal choices of beta and lambda are quite different for different datasets. This increases the complexity in applying the proposed loss in real applications. In the experiments, how did you decide the optimal choices?
3. For the evaluation of Case-2, how is ERM implemented? I think it’s not fair to compare GEM with a vanilla ERM trained on clean data if the test data shift is supposed to be known. Simply data augmentation (e.g., image compression or Gaussian Blur) should work better.
4. For few-shot learning settings, transfer learning is a common technique to address the issue of data limitation. Training from scratch is far from state-of-the-art. It’s interesting to see whether GEM also improves the accuracy when a pre-trained model is utilized for few-shot learning.

**Questions:**

See Weaknesses

---

> ### Author Response · Authors · 2024-11-28
> **Response to Reviewer 41QK (Part 1)**
>
> We thank you very much for taking time to review our paper and provide valuable feedbacks. Below, please find our point-by-point responses to your comments.
>
> > Comment: The theoretical part is a bit tricky to me. Eventually the generalization error is reduced to be two loss terms only relying on test data as shown in Eq (15). Then the algorithm is implemented by simply replacing the test data with training data or processed training data. For the latter case, the difference between training and test data must be known in advance, which is not very realistic. Also, the theoretical result is based on the particular form of squared generalization error, which is different from the common definition of generalization error (training loss - test loss). The motivation of using the squared form of generalization error should be explained.
>
> **Response**:
> - To begin with, we would like to point out an important conceptual distinction between testing data and its probability distribution. While these two concepts are related, they are fundamentally different. Testing data refers to specific data samples (i.e., datasets) used exclusively for evaluation, which must not and should not be used during the training phase. In contrast, the probability distribution of testing data represents the general probabilistic characteristics of the data population, which is typically assumed to be related to, equal to, or inferred from, the probability distribution of the training data. Confusing these concepts can lead to significant misunderstandings and make it difficult to fully grasp the content of our paper. We kindly ask you to keep this distinction in mind as you review our responses to your comments below.
> - Firstly, we would like to clarify a misunderstanding in the comment “the generalization error is reduced to be two loss terms only relying on test data”. In fact, as we mentioned in Line 246~247 of our submission, the generalization error does **not** rely on the test data itself; instead, it only relies on **the distribution of the test data**. Therefore, we consider two cases, where in Case 1 the distribution of the test data is identical to that of the training data (which is typically assumed in most learning tasks), or in Case 2 the distribution of the test data can be inferred from the training data. In Case 1, we can replace the expectation over the pair of random variables $(U, V)$ by the expectation over the pair of random variables $(X, Y)$ since these two quantities are equal given the distributional equivalence. Similarly, we replace the expectation over the pair of random variables $(U, V)$ by the expectation over the pair of random variables $(\hat{X}, Y)$ in Case 2, again, leveraging the distributional equivalence.
> - Secondly, we argue that the Case 2 scenario is quite  realistic. Actually, DNN’s robustness against image compression, or low image quality in general, is a highly practical and widely studied problem [1-2], and a well-know method to improve this robustness is proposed in [2] dubbed stability training (ST). In our revision, we include a comparison with ST in Sec. A.6, and we achieve a clear gain over it. Moreover, we’d like to reiterate that we don’t assume “the difference between training and test data must be known in advance”. In fact, as long as the distribution of the test data can be inferred from the training data by some way, we are able to apply GEM in Case 2, while JPEG compression and Gaussian blurring are just two realistic examples of such a case.
> - Thirdly, there are many different ways to evaluate generalization in the literature, including the one you mentioned. The intuition beind them is clear: to measure DNN’s performance difference between training and testing. The difference between two objects, mathematically, is measured by distance. Therefore, the remaining problem is to determine which distance metric to use. In our case, we choose (the squared) $L^2$ distance since it’s the popular Euclidean distance, and more importantly, it results in a nice bias-variance decomposition.

---

> ### Author Response · Authors · 2024-11-28
> **Response to Reviewer 41QK (Part 2)**
>
> > Comment: The optimal choices of beta and lambda are quite different for different datasets. This increases the complexity in applying the proposed loss in real applications. In the experiments, how did you decide the optimal choices?
>
> **Response**: We’ve made a discussion about this in the Appendix of our original submission. Please refer to Sec. A.3 for more details. In general, as long as $\lambda$ and $\beta$ are appropriately small to avoid slow convergence and gradient explosion at the beginning of training, and $\beta$ is not less than $\lambda$, the experimental results are not sensitive to hyperparameters tuning. For example, for all additional experiments we conducted for GEM during rebuttal (using new CNN and transformer models, or doing transfer learning instead of training from scratch), we have directly reused the same hyperparameters as the ones we reported in our original submission without any fine-tuning. Even with different models and different training settings, the previous hyperparameters are still valid and can maintain the effectiveness of GEM.
>
> > Comment: For the evaluation of Case-2, how is ERM implemented? I think it’s not fair to compare GEM with a vanilla ERM trained on clean data if the test data shift is supposed to be known. Simply data augmentation (e.g., image compression or Gaussian Blur) should work better.
>
> **Response**: Thank you for pointing it out. Integrating your suggestion with Q2 of reviewer fhoH, we decide to add a comparsion with ST in Sec. A.6, where GEM demonstrates better performance than ST. The loss of ST is computed from both the training data $X$ and processed training data $\hat{X}$, thus providing a fair comparison with GEM in Case 2. Please refer to Sec. A.6 of our revision for more details.
>
> > Comment: For few-shot learning settings, transfer learning is a common technique to address the issue of data limitation. Training from scratch is far from state-of-the-art. It’s interesting to see whether GEM also improves the accuracy when a pre-trained model is utilized for few-shot learning.
>
> **Response**: Thank you for this suggestion. We have included more experiments on few-shot transfer learning in Sec. A.10 of our revision. In general, GEM can still achieve some performance gain, especially when the data is more limited. Please refer to Sec. A.10 for more details.
>
> Additionally, please note that our goal is not specifically to improve the performance of few-shot learning. Rather, we use the few-shot learning setting as another potential overfitting example. As mentioned in Line 427~430 of our original submission, GEM consistently outperforms ERM by a large margin, and the performance improvement gradually increases as the training set becomes smaller and smaller, demonstrating the consistent ability of GEM to mitigate overfitting whenever overfitting exists.
>
> [1] Samuel Dodge and Lina Karam. Understanding how image quality affects deep neural networks. In 2016 eighth international conference on quality of multimedia experience (QoMEX), pp. 1–6. IEEE, 2016. \
> [2] Stephan Zheng, Yang Song, Thomas Leung, and Ian Goodfellow. Improving the robustness of deep neural networks via stability training. CVPR 2016.
>
> **We thank you again for your time and effort to review our paper.  If our responses above clear your misunderstandings and address all your concerns, could you kindly increase your score to a higher level.**

---

> ### Author Response · Authors · 2024-12-02
>
> Dear Reviewer 41QK,
>
> Thanks again for reviewing our paper. Since the discussion period is going to end soon, we are eager to know if you are satisfied with our previous response. If no, please kindly tell us your remaining concerns and hopefully we can address them before the deadline. If yes, we wonder if it’s possible for you to raise the score. In any case, we would be extremely grateful to hear from you.
>
> Thanks.

---

### Note · Authors · 2025-01-29

I have read and agree with the venue's withdrawal policy on behalf of myself and my co-authors.